# Defending Backdoor Data Poisoning Attacks Using Noisy Label Defense Algorithm

## Abstract

Training deep neural networks with data corruption is a challenging problem. One example of such corruption is the backdoor data poisoning attack, in which an adversary strategically injects a backdoor trigger to a small fraction of the training data to subtly compromise the training process. Consequently, the trained deep neural network would misclassify testing examples that have been corrupted by the same trigger. While the label of the data could be changed to arbitrary values by an adversary, the extent of corruption injected to the feature values are strictly limited in order to keep the backdoor attack in disguise, which leads to a resemblance between the backdoor attack and a milder attack that involves only noisy labels. In this paper, we investigate an intriguing question: *Can we leverage algorithms that defend against noisy labels corruptions to defend against general backdoor attacks?* We first discuss the limitations of directly using the noisy-label defense algorithms to defend against backdoor attacks. Next, we propose a meta-algorithm that transforms an existing noisy label defense algorithm to one that protects against backdoor attacks. Extensive experiments on different types of backdoor attacks show that, by introducing a lightweight alteration for minimax optimization to the existing noisy-label defense algorithms, the robustness against backdoor attacks can be substantially improved, while the intial form of those algorithms would fail in presence of a backdoor attacks.

## 1 Introduction

Deep neural networks (DNN) have achieved significant success in a variety of applications such as image classification (Krizhevsky et al., 2012), autonomous driving (Major et al., 2019), and natural language processing (Devlin et al., 2018), due to its powerful generalization ability. In the meantime, DNN can be highly susceptible to even small perturbations of training data, which has raised considerable concerns about its trustworthiness (Liu et al., 2020). One representative perturbation approach is backdoor attack, which undermines the DNN performance by modifying a small fraction of the training samples with specific triggers injected into their input features, whose ground-truth labels are altered accordingly to be the attacker-specified ones. It is unlikely to detect backdoor attacks by monitoring the model training performance, since the trained model can still perform well on the benign validation samples. Consequently, during testing phase, if the data is augmented with the trigger, it would be classified as the attacker-specified label. Subtle yet effective, backdoor attacks can pose serious threats to the practical application of DNNs.

Another typical type of data poisoning attack is noisy label attacks (Han et al., 2018; Patrini et al., 2017; Yi & Wu, 2019; Jiang et al., 2017), in which the labels of a small fraction of data are altered deliberately to compromise the model learning, while the input features of the training data remain untouched. Backdoor attacks share a close connection to noisy label attacks, in that during a backdoor attack, the feature can only be altered insignificantly to put the trigger in disguise, which makes the corrupted feature (*e.g.* images with the trigger) highly similar to the uncorrupted ones. Prior efforts have been made to effectively address *noisy label attacks*. For instance, there are algorithms that can tolerate a large fraction of label corruption, with up to 45% noisy labels (Han et al., 2018; Jiang et al., 2018). However, to the best of our knowledge, most algorithms defending against *backdoor attacks* cannot deal with a high corruption ratio even if the features of corrupted data are only slightly perturbed. Observing the limitation of prior arts, we aim to answer one key question: *Can one train a deep neural network that is robust against a large number of backdoor attacks?*

Moreover, given the resemblance between noisy label attacks and backdoor attacks, we also investigate another intriguing question: *Can one leverage algorithms initially designed for handling noisy label attacks to defend against backdoor attacks more effectively?*

The contributions of this paper are multi-fold. First, we provide a novel and principled perspective to decouple the challenges of defending backdoor attacks into two components: one induced by the corrupted input features, and the other induced by the corrupted labels, based on which we can draw a theoretical connection between the noisy-label attacks and backdoor data attacks. Second, we propose a meta-algorithm which addresses both challenges by a novel minimax optimization. Specifically, the proposed approach takes any noisy-label defense algorithm as the input and outputs a reinforced version of the algorithm that is robust against backdoor poisoning attacks, while the initial form of the algorithm fails to provide such defense. Extensive experiments show that the proposed meta-algorithm improves the robustness of DNN models against various backdoor attacks on a variety of benchmark datasets with up to 45% corruption ratio. Furthermore, we propose a systematic, meta-framework to solve backdoor attacks, which can effectively join existing force in noisy label attack defenses and provides more insights to future development of defense algorithms.

## 2 RELATED WORK

### 2.1 ROBUST DEEP LEARNING AGAINST ADVERSARIAL ATTACK

Although DNNs have shown high generalization performance on various tasks, it has been observed that a trained DNN model would yield different results even by perturbing the image in an invisible manner (Goodfellow et al., 2014; Yuan et al., 2019). Prior efforts have been made to tackle this issue, among which one natural defense strategy is to change the empirical loss minimization into a minimax objective. By solving the minimax problem, the model is guaranteed a better worst-case generalization performance (Duchi & Namkoong, 2021). Since exactly solving the inner maximization problem can be computationally prohibitive, different strategies have been proposed to approximate the inner maximization optimization, including heuristic alternative optimization, linear programming Wong & Kolter (2018), semi-definite programming Raghunathan et al. (2018), etc. Besides minimax optimization, another approach to improve model robustness is imposing a Lipschitz constraint on the network. Work along this line includes randomized smoothing Cohen et al. (2019); Salman et al. (2019), spectral normalization Miyato et al. (2018a), and adversarial Lipschitz regularization Terjék (2019). Although there are algorithms that are robust against adversarial samples, they are not designed to confront backdoor attacks, in which clean training data is usually inaccessible. There are also studies that investigated the connection between adversarial robustness and robustness against backdoor attack (Weber et al., 2020). However, to our best knowledge, there is no literature studying the relationship between label flipping attack and backdoor attack.

### 2.2 ROBUST DEEP LEARNING AGAINST LABEL NOISE

Many recent studies have investigated the robustness of classification tasks with noisy labels. For example, Kumar et al. (2010) proposed the Self-Paced Learning (**SPL**) approach, which assigns higher weights to examples with a smaller loss. A similar idea was used in Curriculum Learning (Bengio et al., 2009), in which a model learns on easier examples before moving to the harder ones. Other methods inspired by SPL include learning the data weights (Jiang et al., 2018) and collaborative learning (Han et al., 2018; Yu et al., 2019). An alternative approach to defending noisy label attacks is label correction (Patrini et al., 2017; Li et al., 2017; Yi & Wu, 2019), which attempts to revise the original labels of the data to recover clean labels from corrupted ones. However, since we do not have the knowledge of which data points have been corrupted, it is nontrivial to obtain provable guarantees for label corrections, unless strong assumptions have been made on the corruption type.

### 2.3 DATA POISONING BACKDOOR ATTACK AND ITS DEFENSE

Robust learning against backdoor attacks has been widely studied recently. Gu et al. (2017) showed that even a small patch of perturbation can compromise the generalization performance when data is augmented with a backdoor trigger. Other types of attacks include the blend attacks (Chen et al., 2017), clean label attacks (Turner et al., 2018; Shafahi et al., 2018), latent backdoor attacks (Yao et al., 2019), etc. While there are various types of backdoor attacks, some attack requires that the adversary not only has access to the data but also can has limited control on the training and inference

process. Those attacks include Trojan attacks and blind backdoor attacks (Pang et al., 2020). We refer readers to Pang et al. (2020) for a comprehensive survey on different types of backdoor attacks.

Various defense methods have been proposed to defend against backdoor attacks. One defense category is to remove the corrupted data by using anomaly detection (Tran et al., 2018; Chen et al., 2018). Another category of work is the model inspection (Wang et al., 2019), which aims to inspect and modify the backdoored model to make it robust against the trigger. There are also other methods of tackling the backdoor attacks, such as randomized smoothing (Cohen et al., 2019; Weber et al., 2020), and the median of means (Levine & Feizi, 2020). However, they are either inefficient or cannot defend against backdoor attacks with a large ratio of corrupted data. Some of the above methods also hinge on a clean set of validation data, while in practice, it is unlikely to guarantee the existence of clean validation data, since validation data is usually a subset of training data. To the best of our knowledge, there is no existing backdoor defense algorithm that is motivated from the label corruption perspective.

## 3 PRELIMINARIES

### 3.1 LEARNING WITH NOISY LABELS

In this section, we review some representative approaches for defending noisy-labels. Although the initial forms of these approaches can be vulnerable to backdoor attacks, we show later in the next section that our proposed meta-algorithm can empower them to effectively confront backdoor attacks. Specifically, we look into two types of nosiy-label defending approaches: *filtering-based* approaches and *consistency-based* approaches.

The filtering-based approach is one of the most effective strategies for defending noisy labels, which works by selecting or weighting the training samples based on indicators such as sample losses (Jiang et al., 2017; Han et al., 2018; Jiang et al., 2020) or gradient norms of the loss-layer (Liu et al., 2021). For instance, Jiang et al. (2017) proposed to assign higher probabilities to samples with lower losses to be selected for model training. Consistency-based approach modifies data labels during model training. Specifically, the Bootstrap approach (Reed et al., 2014) encourages model predictions to be consistent between iterations, by modifying the labels as a linear combination of the observed labels and previous predictions.

In this paper, we leverage two filtering-based noisy label algorithms, *i.e.* the Self-Paced Learning (**SPL**) (Jiang et al., 2017; Kumar et al., 2010) and Provable Robust Learning (**PRL**) (Liu et al., 2021), and one consistency-based algorithm, *i.e.* the **Bootstrap** (Reed et al., 2014), to investigate the efficacy of the proposed meta algorithm. Strong empirical results in Section 5 on those input algorithms suggest that our meta framework is readily to benefit other robust noisy-label algorithms. We briefly summarize the main idea of the above algorithms in table 1.

|  | PRL (Filtering-based) | SPL (Filtering-based) | Bootstrap (Modifying Label) |
|---|---|---|---|
| Mini-batch | Keep data with small loss-layer gradient norm | Keep data with small loss | $y_{true} = \alpha y_{true} + (1-\alpha)y_{pred}$ |

Table 1: Overview of noisy-label defending algorithms, which achieve robustness against up to 45% of pairwise flipping label noises.

### 3.2 PROBLEM SETTING OF BACKDOOR ATTACKS

In this paper, we follow a standard setting for backdoor attacks and assume that there is an *adversary* that tries to perform the backdoor attack. Firstly, the adversary can choose up to $\epsilon$ fraction of clean labels $\mathbf{Y} \in \mathbb{R}^{n \times q}$ and modify them to arbitrary valid numbers to form the corrupted labels $\mathbf{Y}_b \in \mathbb{R}^{\lfloor n\epsilon \rfloor \times q}$. Let $\mathbf{Y}_r$ represent the remaining untouched labels, then the final training labels can be denoted as $\mathbf{Y}_\epsilon = [\mathbf{Y}_b, \mathbf{Y}_r]$. Accordingly, the corresponding original feature are denoted as $\mathbf{X} = [\mathbf{X}_o \in \mathbb{R}^{\lfloor n\epsilon \rfloor \times d}, \mathbf{X}_r \in \mathbb{R}^{(n-\lfloor n\epsilon \rfloor) \times d}]$. The adversary can design a trigger $\mathbf{t} \in \mathbb{R}^d$ to form the corrupted feature set $\mathbf{X}_b \in \mathbb{R}^{\lfloor n\epsilon \rfloor \times d}$ such that for any $\mathbf{b}_i$ in $\mathbf{X}_b$, $\mathbf{o}_i$ in $\mathbf{X}_o$, it satisfies $\mathbf{b}_i = \mathbf{o}_i + \mathbf{t}$.

Finally, the training feature will be $\mathbf{X}_\epsilon = [\mathbf{X}_b \in \mathbb{R}^{\lfloor n\epsilon \rfloor \times d}, \mathbf{X}_r \in \mathbb{R}^{(n-\lfloor n\epsilon \rfloor) \times d}]$. Without ambiguity,[1] we also denote $\mathbf{T} = [\mathbf{t}, \mathbf{t}, ..., \mathbf{t}] \in \mathbb{R}^{\lfloor n\epsilon \rfloor \times d}$, so that $\mathbf{X}_o + \mathbf{T} = \mathbf{X}_b$.

Before analyzing the algorithm, we make following assumptions about the adversary attack:

**Assumption 1** (Bounded Corruption Ratio). *The overall corruption ratio and the corruption ratio in each class is bounded. Specifically,*

$$\mathbb{E}_{(\mathbf{x},\mathbf{y},\mathbf{y}_b)\in(\mathbf{X},\mathbf{Y},\mathbf{Y}_b)} \left[ \frac{\mathbf{I}(\mathbf{y}_b = c | \mathbf{y} \neq c)}{\mathbf{I}(\mathbf{y} = c)} \right] \leq \epsilon = 0.5 \, \forall c \in \triangle \mathbf{Y}.$$

**Assumption 2** (Small Trigger). *Any backdoor trigger satisfies $\|\mathbf{t}\|_p \leq \tau$, which subtly alters the data within a small radius-$\tau$ ball without changing its ground-truth label.*

We also assume that there exists at least one black-box robust algorithm $\mathcal{A}$ which can defend any noisy label attacks so long as the noisy-label ratio is bounded by $\epsilon$. Note that the above assumption is mild, since a variety of existing algorithm can handle noisy labels attacks with a large corruption rate (*e.g.* 45%) (Jiang et al., 2017; Han et al., 2018; Reed et al., 2014; Liu et al., 2021).

## 4 METHODOLOGY

Given an $\epsilon$-backdoor attacked dataset $(\mathbf{X}^\epsilon, \mathbf{Y}^\epsilon)$, a clean distribution $p^* := (\mathbf{X}, \mathbf{Y})$, and a loss function $\mathcal{L}$, we aim to answer the following questions: 1) *can we learn a network function $f$ that minimizes the generalization error under the **corrupted** distribution, i.e. $\mathbb{E}_{(x,y)\sim p^*}[\mathcal{L}(f(x+\mathbf{t}),y)]$?* 2) *can the learned $f$ also minimize the generalization error under the ground-truth, **clean** distribution, i.e. $\mathbb{E}_{(x,y)\sim p^*}[\mathcal{L}(f(x),y)]$?* Next, we elaborate our meta-approach for defending against backdoor attacks, which answers the above two questions affirmatively.

### 4.1 A BLACK-BOX ROBUST ALGORITHM AGAINST NOISY LABELS

The ultimate goal for defending against backdoor attacks is to learn a network function $f$ to minimize its risk given corrupted feature inputs:

$$\min_f J(f) := \mathbb{E}_{(x,y)\sim p^*}[\mathcal{L}(f(x+\mathbf{t}),y)]. \tag{1}$$

However, Equation 1 is not directly optimizable due to two-fold challenges: 1) the corrupted *inputs* $\mathbf{X}^\epsilon$ with an unknown trigger $\mathbf{t}$, and 2) the corrupted *labels* $\mathbf{Y}^\epsilon$. That is, neither the clean inputs or ground-truth labels are available. As such, we turn to an approachable objective that optimizes the worst-case of Equation 1:

$$\min_f \max_{\|\mathbf{c}\|_p \leq \tau} \frac{1}{n} \sum_{\mathbf{x}\in\mathbf{X},\mathbf{y}\in\mathbf{Y}} [\mathcal{L}(f(\mathbf{x}+\mathbf{c}),\mathbf{y})]. \tag{2}$$

Since the trigger satisfies $\|\mathbf{t}\|_p \leq \tau$, it is easy to see that Equation 2 minimizes an *upper-bound* of the ground-truth loss, in that:

$$\frac{1}{n} \sum_{\mathbf{x}\in\mathbf{X},\mathbf{y}\in\mathbf{Y}} \mathcal{L}(f(\mathbf{x}+\mathbf{t}),\mathbf{y}) \leq \max_{\|\mathbf{c}\|_p \leq \tau} \frac{1}{n} \sum_{\mathbf{x}\in\mathbf{X},\mathbf{y}\in\mathbf{Y}} [\mathcal{L}(f(\mathbf{x}+\mathbf{c}),\mathbf{y})].$$

To this end, directly optimizing the surrogate objective in Equation 2 is still intractable, since we do not have access to clean $\mathbf{X}$ and $\mathbf{Y}$, which are involved in the inner maximization loop. To tackle this challenge, we will first assume that the clean label $\mathbf{Y}$ is available, and then relax this prerequisite by using any learning algorithms that are robust against noisy labels. Specifically, assume that $\phi_{\mathbf{w}} = \mathcal{L} \circ f$ has a Lipschitz constant $L$ *w.r.t* $\mathbf{x}$, we further obtain a new upper bound (see Appendix for derivation details):

$$\frac{1}{n} \sum_{\mathbf{x}\in\mathbf{X},\mathbf{y}\in\mathbf{Y}} [\mathcal{L}(f(\mathbf{x}+\mathbf{c}),\mathbf{y})] \leq \frac{1}{n} \sum_{\mathbf{x}\in\mathbf{X}^\epsilon,\mathbf{y}\in\mathbf{Y}} \phi_w(\mathbf{x}_i+c,\mathbf{y}) + \epsilon\tau L, \tag{3}$$

---

[1] Some backdoor attack algorithms design instance-specific trigger. In this paper, we only focus on the static trigger case and leave the instance-specific trigger case for our future study.

which draws a principled connection between the risks when using corrupted data and clean data:

$$\min_f \max_{\|\mathbf{c}\|_p \leq \tau} \frac{1}{n} \sum_{\mathbf{x} \in \mathbf{X}, \mathbf{y} \in \mathbf{Y}} [\mathcal{L}(f(\mathbf{x} + \mathbf{c}), \mathbf{y})] \approx \left\{ \min_f \max_{\|\mathbf{c}\|_p \leq \tau} \frac{1}{n} \sum_{\mathbf{x} \in \mathbf{X}^\epsilon, \mathbf{y} \in \mathbf{Y}} [\mathcal{L}(f(\mathbf{x} + \mathbf{c}), \mathbf{y})] + \epsilon \tau L \right\}, \quad (4)$$

where the first term on the RHS of Equation 4 involves optimization on the *corrupted* features $\mathbf{X}^\epsilon$ and *clean* labels $\mathbf{Y}$, while the second term on the RHS requires minimizing the Lipschitz constant $L$ w.r.t $\mathbf{x}$. Recall that minimizing the maximum gradient norm is equivalent to minimizing the Lipschitz constant (Terjék, 2019). Therefore, optimizing the first term naturally regulates the maximum change of the loss function within a small ball, which hence constrains the magnitude of the gradient and has negligible effects on the Lipschitz regularization. The relationship between Lipschitz regularization and adversarial training has also been well discussed by literatures (Terjék, 2019; Miyato et al., 2018b). We defer more discussion along this line to Appendix.

Equation 4 indicates that if the target labels are not corrupted, and the learned function has a small Lipschitz constant, learning with corrupted features is feasible to achieve low risks. Up to now, the remaining challenge of optimizing the surrogate objective in Equation 4 is the inaccessible clean label set $\mathbf{Y}$. Fortunately, a variety of algorithms are at hand for handling noisy labels during learning (Jiang et al., 2017; Liu et al., 2021; Kumar et al., 2010), which we can directly apply to our minimax optimization scheme. Specifically, for the outer minimization, one can have:

$$\min_f \frac{1}{n} \sum_{\mathbf{x} \in \mathbf{X}^\epsilon, \mathbf{y} \in \mathbf{Y}^\epsilon} [\mathcal{L}(f(\mathbf{x} + \mathbf{c}), \mathbf{y})],$$

For the outer minimization, we can perform the noisy-label update for the above optimization objective. For instance, Given the mini-batch $\mathbf{M}_x$, $\mathbf{M}_y$ with batch size $m$, if we use SPL to perform the update, we can get the top $(1 - \epsilon)m$ data with a small risk $\mathcal{L}(f(\mathbf{x} + \mathbf{c}), \mathbf{y})$ to perform one-step gradient descent. If we use the PRL to perform the update, assume $\mathcal{L}$ is the cross-entropy loss, one can get the top $(1-\epsilon)m$ data with small loss-layer gradient norm $\|f(\mathbf{x}+\mathbf{c})-\mathbf{y}\|$ to perform one-step gradient descent. If we use the bootstrap, we can directly add a bootstrap regularization to update the above objective.

In the meantime, it is non-trivial to directly solve the inner maximization, since adversarial learning $\mathbf{c}$ in Equation 4 still faces the threat of noisy labels. To tackle this issue, we can leverage the same robust noisy label algorithm. Specifically, we first approximate the inner optimization using the first-order Tyler expansion:

$$\mathbf{c}^* = \arg\max_{\|\mathbf{c}\|_p \leq \tau} \frac{1}{n} \sum_{\mathbf{x} \in \mathbf{X}, \mathbf{y} \in \mathbf{Y}} \mathcal{L}(f(\mathbf{x} + \mathbf{c}), \mathbf{y}) \approx \arg\max_{\|\mathbf{c}\|_p \leq \tau} \frac{1}{n} \sum_{\mathbf{x} \in \mathbf{X}, \mathbf{y} \in \mathbf{Y}} \mathcal{L}(f(\mathbf{x}), \mathbf{y}) + \mathbf{c}^T \nabla_\mathbf{x} \mathcal{L}(f(\mathbf{x}), \mathbf{y})$$

$$= \arg\max_{\|\mathbf{c}\|_p \leq \tau} \mathbf{c}^T \nabla_\mathbf{x} \frac{1}{n} \sum_{\mathbf{x} \in \mathbf{X}, \mathbf{y} \in \mathbf{Y}} \mathcal{L}(f(\mathbf{x}), \mathbf{y}).$$

Above optimization is a linear programming problem. With the $l_\infty$ norm ball constraint on the perturbation, the above update can be reduced to the fast gradient sign method (FGSM). Given a minibatch $\mathbf{M}_x$, $\mathbf{M}_y$ with batchsize $m$, we can have the closed-form solution as the following:

$$\tilde{\mathbf{c}} = \text{Clip}_\mathbf{c} \left\{ \frac{\tau}{m} \cdot \sum_{\mathbf{x} \in \mathbf{M}_x, \mathbf{y} \in \mathbf{M}_y} \text{sign}\left( \nabla_\mathbf{x} \mathcal{L}\left( f(\mathbf{x}), \mathbf{y} \right) \right) \right\}. \quad (5)$$

To relax the prerequisite of a clean label set $\mathbf{y}$ in Equation 5, we will use a noisy-label algorithm to perform the update. For instance, if we use a loss-filtering based algorithm (*e.g.* SPL), then for each mini-batch, only the top $(1 - \epsilon)m$ data with small $\mathcal{L}\left( f(\mathbf{x}), \mathbf{y} \right)$ would be included in the update. If we adopt a gradient-based filtering algorithm (*e.g.* PRL), given that $\mathcal{L}$ is the cross-entropy loss, then only top $(1-\epsilon)m$ data with small $\|f(\mathbf{x})-\mathbf{y}\|$ will be included. The outside clipping ensures that the feature value of the corrupted image is in the valid range. Based on the above building blocks, we now introduce our algorithm in Algorithm 1, a meta defense scheme that are robust against backdoor attacks, given arbitrary noisy-label robust algorithm $\mathcal{A}$ as an input. The algorithm is illustrated in Figure 1.

---

**Algorithm 1:** Meta algorithm for Robust Learning Against Backdoor Attacks

---

**input:** Corrupted training data $\mathbf{X}^\epsilon, \mathbf{Y}^\epsilon$, perturbation limit: $\tau$, learning with noisy label algorithm $\mathcal{A}$ (*e.g.* PRL, SPL, Bootstrap).
**return** *trained neural network* ;
**while** *epoch $\leq$ max_epoch* **do**

    **for** *sampled minibatch* $\mathbf{M}_x, \mathbf{M}_y$ *in* $\mathbf{X}^\epsilon, \mathbf{Y}^\epsilon$ **do**

        #Inner maximization step

        initialize $\mathbf{c}$ as 0 vector.

        optimize the objective $\max_{\|\mathbf{c}\| \leq \tau} \mathcal{L}(f(\mathbf{M}_x + \mathbf{c}), \mathbf{M}_y)$ *w.r.t* to $\mathbf{c}$ by using robust algorithm $\mathcal{A}$

        optimize the objective $\min_f \mathcal{L}(f(\mathbf{M}_x + \mathbf{c}), \mathbf{M}_y)$ *w.r.t* $f$ by using robust algorithm $\mathcal{A}$

    **end**

**end**

---

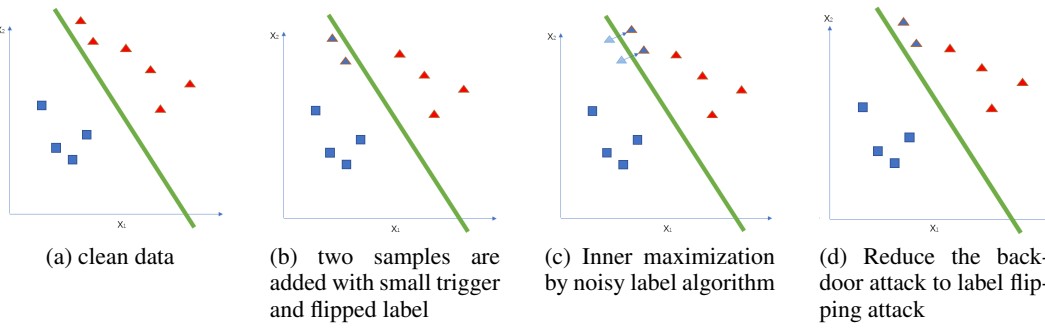

(a) clean data      (b) two samples are added with small trigger and flipped label      (c) Inner maximization by noisy label algorithm      (d) Reduce the backdoor attack to label flipping attack

Figure 1: Illustration of our meta algorithm. By combining the minimax objective and noisy label algorithm, we could reduce the backdoor attack problem to the label flipping attack. The left most is the clean original data and the second one is corrupted data. The third figure shows the inner maximization step while the last figure shows the outer minimization step.

## 4.2 THEORETICAL JUSTIFICATION

Our ultimate goal is to learn $\mathbf{w}$ that achieves a small risk $\mathbb{E}_{\mathbf{x}, \mathbf{y} \sim p*} \phi_\mathbf{w}(\mathbf{x} + \mathbf{t}, \mathbf{y})$. To study the generalization performance on the ground-truth distribution $p*$, we first define the following risks:
$\mathcal{R}_t^{emp} = \frac{1}{n} \sum_{\mathbf{x} \in \mathbf{X}, \mathbf{y} \in \mathbf{Y}} \phi_\mathbf{w}(\mathbf{x} + \mathbf{t}, \mathbf{y})$, $\mathcal{R}_t = \mathbb{E}_{\mathbf{x}, \mathbf{y} \sim p*} \phi_\mathbf{w}(\mathbf{x} + \mathbf{t}, \mathbf{y})$, $\mathcal{R}_c^{emp} = \frac{1}{n} \sum_{\mathbf{x} \in \mathbf{X}^\epsilon, \mathbf{y} \in \mathbf{Y}} \phi_\mathbf{w}(\mathbf{x} + \mathbf{c}, \mathbf{y})$, and $\tilde{\mathcal{R}}_c^{emp} = \frac{1}{n} \sum_{\mathbf{x} \in \mathbf{X}^\epsilon, \mathbf{y} \in \mathbf{Y}^\epsilon} \phi_\mathbf{w}(\mathbf{x} + \mathbf{c}, \mathbf{y})$. Next, we focus on the gap between $\mathcal{R}_t$ and $\mathcal{R}_c^{emp}$.

**Theorem 1.** *Let $\tilde{\mathcal{R}}_c^{emp}, \mathcal{R}_c^{emp}, \mathcal{R}_t, \epsilon, \tau$ defined as above. Assume that the prior distribution of the network parameter $\mathbf{w}$ is $\mathcal{N}(0, \sigma)$, and the posterior distribution of parameter is $\mathcal{N}(\mathbf{w}, \sigma)$ which is learned from the training data. Let $k$ be the number of parameters, and $n$ be the sample size. If the objective function $\phi_\mathbf{w} = \mathcal{L} \circ f$ is $L_\phi$-Lipschitz smooth, then with probability at least 1-$\delta$, one can derive:*

$$\mathcal{R}_t \leq \mathcal{R}_c^{emp} + L_\phi(2\tau + \epsilon\tau) + \sqrt{\frac{\frac{1}{4} k \log\left(1 + \frac{\|\boldsymbol{w}\|_2^2}{k\sigma^2}\right) + \frac{1}{4} + \log \frac{n}{\delta} + 2\log(6n + 3k)}{n - 1}}. \tag{6}$$

We hereby present the skeleton of the proof and defer more details to Appendix. First, we decompose the error into two components: 1) the generalization gap on the triggered data, and 2) the difference of performance loss between the trigger $\mathbf{t}$ and worst case perturbation $\mathbf{c}$: $\mathcal{R}_t - \mathcal{R}_c^{emp} = (\mathcal{R}_t - \mathcal{R}_t^{emp}) + (\mathcal{R}_t^{emp} - \mathcal{R}_c^{emp})$.

The first components is $\sqrt{\frac{\frac{1}{4} k \log\left(1 + \frac{\|\boldsymbol{w}\|_2^2}{k\sigma^2}\right) + \frac{1}{4} + \log \frac{n}{\delta} + 2\log(6n + 3k)}{n - 1}}$, which is derived by following the PAC-Bayes framework (Foret et al., 2020). For the second term, the gap is introduced by two sources. The first source is the difference between $\mathbf{c}$ and $\mathbf{t}$, and the second is from the difference between $\mathbf{X}$ and $\mathbf{X}^\epsilon$. Since the objective is $L_\phi$ Lipschitz, and $\|\mathbf{t} - \mathbf{c}\| \leq 2\tau$ according to our constraint to the adversary, it is easy to upper bound the error as $2\tau L_\phi$. In the meantime, there is $\epsilon$-fraction of

difference between $\mathbf{X}$ and $\mathbf{X}^\epsilon$, which is bounded by $\|\mathbf{t}\| < \tau$ and leads to the other difference term $L_\phi \epsilon \tau$.

Theorem 1 presents an upper-bound of the gap $\mathcal{R}_t - \mathcal{R}_c^{emp}$. The first term in Equation 6 can be minimized by using a noisy label algorithm. The second term, which is the error induced by the adversarial trigger, is jointly constrained by the Lipschitz constant $L_\phi$, perturbation limit $\tau$, and the corruption ratio $\epsilon$. We can regularize the $L_\phi$ whereas the $\tau$ and $\epsilon$ are controlled by the unknown adversary. Note that existing literature has also shown that adversarial training plays a resemblant role as the Lipschitz regularization. The last term, *i.e.* the normal generalization error on the clean data, is difficult to minimize directly. The bound in Theorem 1 emphasizes the importance of involving both the noisy label algorithm and the adversarial training. The noisy label algorithm can reduce the $\mathcal{R}_c^{emp}$ while the adversarial training minimizes the Lipschitz constant $\mathcal{L}_\phi$ to optimize the second term.

## 5 EXPERIMENT

In this section, we perform the empirical study. In our experiment, we perform experiment on CIFAR10 and CIFAR100 benchmark data. We use ResNet-32 (He et al., 2016) as the backbone network structure for all experiment baseline. The initial learning rate of all methods are set to be 3e-4, and we use AdamW (Loshchilov & Hutter, 2017) as the optimizer for all methods. The evaluation metric is the top-1 accuracy for both clean testing data and testing data with backdoor trigger.

For the backdoor data poisoning attack, we use simple badnet attack (Gu et al., 2017) and gaussian blending attack (Chen et al., 2017) since these two attacks do not require any information about the model or training procedure (Pang et al., 2020).

- *badnet patch attack*: trigger is a $3 \times 3$ black-white checkerboard and it is added to the right bottom corner of the image.
- *blending attack*: trigger is a fixed Gaussian noise which has the same dimension as the image. The corrupted image generated by $\mathbf{x}_i^\epsilon = (1 - \alpha)\mathbf{x}_i + \alpha \mathbf{t}$. In our experiment, we set the $\alpha$ as 0.1.

The poisoned sample can be found at Appendix. We deploy the multi-target backdoor attack in this paper. Our poisoning approach is as following: we first systematically flip the label to perform the label-flipping attack . Then, for those being attacked samples, we add the triggers on their feature. Without adding the trigger, the problem would reduce to the noisy label problem.

and we use the following two evaluation metric.

- top-1 clean accuracy: the top-1 accuracy calculated on trigger-less testing data
- top-1 poison accuracy: the top-1 accuracy calculated by the model prediction on poisoned testing data and ground truth clean label.

The first metric describes how model performs on benign data while the second metric describes how model performs on triggered data. We varies our training data poisoning rate as $[15\%, 25\%, 35\%, 45\%]$ to investigate how our algorithm performs against different corruption ratio. All methods are trained 100 epochs, Also, in this paper, we assume there is *no clean validation data available*. Thus, it is difficult to perform early stopping or decide which epoch result should be used. Thus, we report the averaged accuracy across last 10 epochs for every methods.

We investigate three noisy label algorithms by comparing the performance of the original method and reinforced method. Specifically, we choose SPL, PRL, and Bootstrap as our original noisy label algorithm and the corresponding reinforced algorithm with adversarial training SPL-AT, PRL-AT, Bootstrap-AT. We also compared our method against adversarial training only (AT), which only uses the adversarial training without using any noisy label algorithm. To show the success of the attack, we also includes the standard training results.

### 5.1 HOW ROBUST LEARNING AGAINST NOISY LABEL PERFORMS ON BACKDOOR DATA

In this section, we aim to answer the first question: *how does noisy label defense algorithm performs against backdoor attack?*

To answer above question, We evaluate PRL, SPL, and Bootstrap on CIFAR10 and CIFAR100 dataset. The results can be found in table 2 and table 3. As we can see in the table, the standard training performs well on the benign testing data (i.e. high clean accuracy) while its performance on triggered data is very bad (i.e. low poison accuracy), which indicates the effectiveness of the backdoor attack.

As for the three noisy label algorithm, we found that although Bootstrap, SPL, and PRL all performs well for the benign testing data, they all fail when defending the backdoor attack data especially with large corruption rate, which illustrates that noisy label algorithm cannot defend the backdoor attack especially when corruption ratio is large.

## 5.2 How adversarial training improves the noisy label algorithm

To investigate whether adversarial training could improve the robustness of existing noisy label algorithm against backdoor attack, we performed experiment on SPL-AT, PRL-AT and Bootstrap-AT to see how they performs on both clean test data and triggered test data. The results can be found in table 2 and table 3. As we can see that by adding the adversarial training, the performance on triggered data is largely improved (i.e. the poison accuracy significantly improves). Also, we noticed that this improvement pattern is hold for all three noisy label algorithm, which indicates the effectiveness of the proposed method on improving robustness against backdoor attack. Also, compared to adversarial training only (AT), adding noisy label does improve the performance. Especially for the PRL algorithm, we see the PRL-AT achieves better performance than AT. Also, we found that compared to consistency based noisy label algorithm, filtering based algorithm are more easier to be boosted by adversarial training. The potential reason behind this could be that filtering based method is more efficient against noisy label algorithm Han et al. (2018); Jiang et al. (2017); Liu et al. (2021).

| | | **Backdoor Attack Defense Accuracy.** | | | | | | | |
|---|---|---|---|---|---|---|---|---|---|
| Dataset | $\epsilon$ | AT | BootStrap | Bootstrap-AT | PRL | PRL-AT | SPL | SPL-AT | Standard |
| CIFAR10 with **Patch** Attack, **Poison** Accuracy | 0.15 | 66.64 ± 5.28 | 2.09 ± 0.13 | 3.05 ± 0.47 | 81.71 ± 0.37 | 80.15 ± 0.42 | 34.60 ± 1.57 | 77.60 ± 3.81 | 2.10 ± 0.10 |
| | 0.25 | 63.98 ± 7.16 | 2.01 ± 0.23 | 2.75 ± 0.17 | 45.94 ± 25.19 | 78.14 ± 0.48 | 10.87 ± 2.13 | 22.17 ± 10.51 | 2.13 ± 0.15 |
| | 0.35 | 60.19 ± 1.35 | 1.98 ± 0.15 | 2.66 ± 0.16 | 31.27 ± 17.63 | 75.04 ± 0.29 | 11.74 ± 1.24 | 15.40 ± 7.56 | 2.01 ± 0.09 |
| | 0.45 | 51.25 ± 1.81 | 1.94 ± 0.12 | 2.53 ± 0.20 | 17.50 ± 1.66 | 58.90 ± 12.52 | 12.32 ± 1.20 | 14.00 ± 5.35 | 1.88 ± 0.04 |
| CIFAR10 with **Patch** Attack, **Clean** Accuracy | 0.15 | 66.77 ± 5.17 | 85.22 ± 0.48 | 82.62 ± 0.26 | 82.06 ± 0.16 | 80.25 ± 0.43 | 77.35 ± 2.76 | 77.70 ± 3.78 | 85.40 ± 0.37 |
| | 0.25 | 63.98 ± 7.16 | 85.25 ± 0.19 | 81.90 ± 0.25 | 78.57 ± 1.03 | 78.22 ± 0.56 | 69.52 ± 2.38 | 68.49 ± 2.76 | 85.20 ± 0.26 |
| | 0.35 | 60.31 ± 1.37 | 84.86 ± 0.13 | 81.75 ± 0.25 | 73.63 ± 0.75 | 75.10 ± 0.31 | 60.23 ± 3.14 | 58.88 ± 3.46 | 84.73 ± 0.13 |
| | 0.45 | 51.25 ± 1.81 | 1.94 ± 0.12 | 2.53 ± 0.20 | 17.50 ± 1.66 | 58.90 ± 12.52 | 50.82 ± 1.48 | 14.00 ± 5.35 | 1.88 ± 0.04 |
| CIFAR10 with **Blend** Attack, **Poison** Accuracy | 0.15 | 65.15 ± 0.94 | 2.17 ± 0.17 | 24.98 ± 10.01 | 6.41 ± 3.91 | 79.71 ± 0.33 | 11.60 ± 6.56 | 74.77 ± 3.53 | 2.29 ± 0.10 |
| | 0.25 | 56.98 ± 0.72 | 2.06 ± 0.10 | 33.33 ± 20.03 | 6.77 ± 2.81 | 76.99 ± 0.37 | 11.60 ± 8.59 | 52.36 ± 10.57 | 2.03 ± 0.18 |
| | 0.35 | 47.84 ± 1.49 | 1.86 ± 0.07 | 13.13 ± 7.11 | 9.42 ± 5.28 | 73.17 ± 0.96 | 12.71 ± 9.33 | 50.79 ± 7.92 | 1.97 ± 0.07 |
| | 0.45 | 34.66 ± 1.49 | 1.83 ± 0.11 | 6.12 ± 2.86 | 8.13 ± 4.50 | 49.88 ± 8.43 | 8.69 ± 4.41 | 35.06 ± 4.00 | 1.88 ± 0.06 |
| CIFAR10 with **Blend** Attack, **Clean** Accuracy | 0.15 | 66.14 ± 0.98 | 85.54 ± 0.58 | 81.44 ± 0.58 | 77.51 ± 1.20 | 80.06 ± 0.34 | 76.25 ± 2.78 | 75.65 ± 3.11 | 85.28 ± 0.34 |
| | 0.25 | 58.91 ± 5.70 | 84.95 ± 0.30 | 80.89 ± 0.65 | 71.45 ± 1.40 | 77.82 ± 0.26 | 67.86 ± 2.58 | 65.08 ± 0.82 | 85.06 ± 0.39 |
| | 0.35 | 50.07 ± 13.26 | 84.72 ± 0.58 | 80.63 ± 0.57 | 66.22 ± 1.15 | 74.34 ± 1.01 | 60.52 ± 2.26 | 60.16 ± 2.39 | 84.72 ± 0.28 |
| | 0.45 | 38.03 ± 15.42 | 84.36 ± 0.38 | 80.35 ± 0.39 | 55.78 ± 2.09 | 57.17 ± 9.02 | 49.48 ± 2.19 | 46.74 ± 0.71 | 84.07 ± 0.17 |

Table 2: Performance on CIFAR10. $\epsilon$ is the corruption rate.

## 5.3 Ablation Study

In this section, we aim to explore more about the proposed framework. Since our algorithm use the noisy-label solver for both inner and outer optimization. A interesting question to ask is that whether both inner and outer noisy-label solver plays an important role in defense the backdoor attack. Thus, we have two variants. One is we only use noisy label algorithm to update the model for outer minimization and another one is we only use the noisy label algorithm to update the model for inner maximization. The results can be found at table 4 and table 5. As we could see in these two tables, using noisy label algorithm to perform the inner maximization is more important compared to using noisy label algorithm to perform out minimization.

## 6 Conclusion

In this paper, we investigate the connection between label flipping attack and backdoor data poisoning attack. We show that although robust algorithm against label flipping attack cannot defend the

| | | | | | Backdoor Attack Defense Accuracy. | | | | |
|---|---|---|---|---|---|---|---|---|---|
| Dataset | $\epsilon$ | AT | BootStrap | Bootstrap-AT | PRL | PRL-AT | SPL | SPL-AT | Standard |
| CIFAR100 with **Patch** Attack, **Poison** Accuracy | 0.15 | 23.70 ± 1.39 | 5.23 ± 0.81 | 44.74 ± 4.05 | 15.15 ± 9.17 | 47.11 ± 0.58 | 24.87 ± 5.27 | 42.24 ± 0.76 | 5.28 ± 0.50 |
| | 0.25 | 21.84 ± 1.17 | 3.07 ± 0.23 | 44.09 ± 1.10 | 17.53 ± 18.06 | 43.81 ± 0.41 | 8.48 ± 1.13 | 35.46 ± 1.13 | 3.10 ± 0.60 |
| | 0.35 | 17.16 ± 1.09 | 2.85 ± 0.12 | 40.14 ± 0.20 | 20.83 ± 10.03 | 39.76 ± 0.72 | 7.37 ± 0.59 | 28.41 ± 1.72 | 3.24 ± 1.04 |
| | 0.45 | 13.61 ± 0.74 | 10.60 ± 10.49 | 31.21 ± 0.30 | 23.98 ± 9.32 | 29.76 ± 1.11 | 7.26 ± 0.76 | 20.43 ± 1.69 | 10.51 ± 11.21 |
| CIFAR100 with **Patch** Attack, **Clean** Accuracy | 0.15 | 34.08 ± 0.40 | 52.39 ± 0.38 | 47.76 ± 0.14 | 50.50 ± 0.41 | 47.21 ± 0.56 | 46.38 ± 0.41 | 42.38 ± 0.73 | 52.42 ± 0.59 |
| | 0.25 | 31.72 ± 0.75 | 50.54 ± 0.25 | 44.82 ± 0.52 | 47.49 ± 0.91 | 43.89 ± 0.35 | 39.98 ± 0.80 | 35.65 ± 1.14 | 50.53 ± 0.55 |
| | 0.35 | 29.50 ± 1.73 | 48.41 ± 0.42 | 40.38 ± 0.18 | 44.21 ± 0.21 | 39.80 ± 0.67 | 34.11 ± 1.10 | 28.52 ± 1.70 | 48.75 ± 0.71 |
| | 0.45 | 23.93 ± 3.43 | 41.46 ± 5.00 | 31.48 ± 0.38 | 34.34 ± 0.91 | 29.79 ± 1.13 | 27.87 ± 2.28 | 20.55 ± 1.75 | 41.02 ± 6.06 |
| CIFAR100 with **Blend** Attack, **Poison** Accuracy | 0.15 | 33.65 ± 0.54 | 2.19 ± 0.28 | 46.65 ± 0.33 | 2.10 ± 0.43 | 46.01 ± 0.50 | 6.14 ± 1.12 | 41.57 ± 0.74 | 2.09 ± 0.20 |
| | 0.25 | 30.95 ± 0.42 | 1.17 ± 0.08 | 41.84 ± 0.59 | 1.45 ± 0.21 | 41.78 ± 0.76 | 2.95 ± 0.56 | 33.54 ± 1.76 | 1.12 ± 0.20 |
| | 0.35 | 27.30 ± 0.45 | 1.05 ± 0.06 | 31.88 ± 1.26 | 1.51 ± 0.17 | 34.51 ± 1.60 | 2.00 ± 0.49 | 25.71 ± 2.31 | 1.08 ± 0.16 |
| | 0.45 | 20.79 ± 4.97 | 0.99 ± 0.07 | 23.61 ± 1.07 | 2.68 ± 1.17 | 22.00 ± 1.95 | 2.39 ± 0.17 | 18.62 ± 1.21 | 0.92 ± 0.11 |
| CIFAR100 with **Blend** Attack, **Clean** Accuracy | 0.15 | 34.22 ± 0.58 | 52.65 ± 0.19 | 47.77 ± 0.36 | 48.61 ± 0.18 | 46.92 ± 0.47 | 46.01 ± 0.40 | 42.40 ± 0.70 | 52.60 ± 0.59 |
| | 0.25 | 33.65 ± 0.55 | 51.12 ± 0.37 | 44.75 ± 0.45 | 45.23 ± 0.34 | 42.87 ± 0.72 | 40.47 ± 1.47 | 35.71 ± 1.10 | 50.98 ± 0.43 |
| | 0.35 | 28.14 ± 0.48 | 49.80 ± 0.24 | 40.85 ± 0.37 | 40.46 ± 0.17 | 36.30 ± 1.24 | 35.70 ± 1.68 | 28.56 ± 2.05 | 49.65 ± 0.49 |
| | 0.45 | 22.03 ± 0.49 | 48.46 ± 0.53 | 34.78 ± 1.39 | 34.98 ± 0.83 | 24.71 ± 1.37 | 29.91 ± 1.40 | 21.82 ± 1.21 | 48.07 ± 0.52 |

Table 3: Performance on CIFAR100. $\epsilon$ is the corruption rate.

| | | | Backdoor Attack Defense Accuracy. | | | | |
|---|---|---|---|---|---|---|---|
| Dataset | $\epsilon$ | BootStrap-inner | Bootstrap-outer | PRL-inner | PRL-outer | SPL-inner | SPL-outer |
| CIFAR10 with **Patch** Attack, **Poison** Accuracy | 0.15 | 3.15 ± 0.61 | 3.20 ± 0.63 | 80.78 ± 0.31 | 2.84 ± 0.23 | 65.50 ± 16.22 | 3.09 ± 0.35 |
| | 0.25 | 2.74 ± 0.10 | 2.73 ± 0.09 | 79.07 ± 0.20 | 2.50 ± 0.10 | 18.95 ± 9.91 | 2.58 ± 0.19 |
| | 0.35 | 2.70 ± 0.24 | 2.67 ± 0.15 | 76.06 ± 0.37 | 2.39 ± 0.26 | 13.45 ± 5.40 | 2.38 ± 0.14 |
| | 0.45 | 2.32 ± 0.08 | 2.51 ± 0.11 | 67.87 ± 2.63 | 2.24 ± 0.10 | 12.10 ± 4.46 | 2.23 ± 0.26 |
| CIFAR10 with **Patch** Attack, **Clean** Accuracy | 0.15 | 82.58 ± 0.33 | 82.45 ± 0.25 | 80.86 ± 0.31 | 83.09 ± 0.12 | 76.48 ± 3.03 | 83.02 ± 0.49 |
| | 0.25 | 82.14 ± 0.28 | 81.87 ± 0.23 | 79.10 ± 0.17 | 83.13 ± 0.21 | 69.33 ± 2.57 | 83.30 ± 0.13 |
| | 0.35 | 81.71 ± 0.46 | 81.55 ± 0.54 | 76.08 ± 0.34 | 82.83 ± 0.38 | 59.76 ± 3.59 | 83.05 ± 0.38 |
| | 0.45 | 81.53 ± 0.16 | 81.00 ± 0.47 | 69.96 ± 0.37 | 82.78 ± 0.18 | 49.31 ± 0.53 | 82.84 ± 0.27 |
| CIFAR10 with **Blend** Attack, **Poison** Accuracy | 0.15 | 29.85 ± 10.65 | 40.79 ± 13.27 | 80.38 ± 0.15 | 46.29 ± 18.09 | 72.89 ± 6.14 | 48.21 ± 14.90 |
| | 0.25 | 14.81 ± 10.42 | 27.57 ± 10.93 | 78.44 ± 0.19 | 27.34 ± 18.42 | 54.46 ± 10.45 | 21.18 ± 11.85 |
| | 0.35 | 6.52 ± 3.80 | 17.41 ± 10.13 | 71.93 ± 2.69 | 11.25 ± 5.92 | 46.12 ± 13.77 | 14.58 ± 7.12 |
| | 0.45 | 11.58 ± 14.94 | 9.01 ± 4.66 | 64.98 ± 2.74 | 5.90 ± 2.28 | 42.30 ± 5.85 | 5.16 ± 1.64 |
| CIFAR10 with **Blend** Attack, **Clean** Accuracy | 0.15 | 81.54 ± 0.25 | 81.18 ± 0.75 | 80.73 ± 0.18 | 82.51 ± 0.36 | 76.11 ± 3.32 | 82.35 ± 0.22 |
| | 0.25 | 80.99 ± 1.12 | 80.37 ± 0.94 | 78.23 ± 0.46 | 82.35 ± 0.65 | 66.64 ± 2.31 | 82.15 ± 0.37 |
| | 0.35 | 81.04 ± 0.81 | 79.54 ± 1.32 | 71.62 ± 2.65 | 82.55 ± 0.47 | 57.44 ± 1.78 | 81.81 ± 1.00 |
| | 0.45 | 81.06 ± 0.25 | 78.93 ± 0.84 | 62.34 ± 2.51 | 82.15 ± 0.48 | 48.82 ± 0.94 | 81.81 ± 1.06 |

Table 4: Ablation study on CIFAR10. $\epsilon$ is the corruption rate.

| | | | Backdoor Attack Defense Accuracy. | | | | |
|---|---|---|---|---|---|---|---|
| Dataset | $\epsilon$ | BootStrap-inner | Bootstrap-outer | PRL-inner | PRL-outer | SPL-inner | SPL-outer |
| CIFAR100 with **Patch** Attack, **Poison** Accuracy | 0.15 | 45.76 ± 2.65 | 41.66 ± 8.37 | 47.34 ± 0.44 | 44.32 ± 5.45 | 43.05 ± 0.39 | 43.41 ± 6.36 |
| | 0.25 | 44.41 ± 1.55 | 43.65 ± 0.88 | 44.71 ± 0.40 | 41.13 ± 5.82 | 35.69 ± 1.05 | 41.81 ± 3.89 |
| | 0.35 | 40.02 ± 0.19 | 38.72 ± 0.60 | 40.19 ± 0.39 | 38.75 ± 0.60 | 24.98 ± 6.34 | 38.97 ± 1.10 |
| | 0.45 | 31.12 ± 0.40 | 29.46 ± 0.33 | 31.49 ± 1.04 | 29.74 ± 0.35 | 20.13 ± 1.80 | 30.19 ± 0.31 |
| CIFAR100 with **Patch** Attack, **Clean** Accuracy | 0.15 | 48.01 ± 0.28 | 47.91 ± 0.21 | 47.43 ± 0.43 | 48.41 ± 0.40 | 43.29 ± 0.21 | 48.12 ± 0.36 |
| | 0.25 | 45.27 ± 0.82 | 44.68 ± 0.27 | 44.83 ± 0.38 | 45.58 ± 0.39 | 36.21 ± 0.80 | 45.14 ± 0.64 |
| | 0.35 | 40.49 ± 0.23 | 38.98 ± 0.47 | 40.40 ± 0.41 | 39.46 ± 0.30 | 29.58 ± 1.49 | 39.89 ± 0.50 |
| | 0.45 | 31.32 ± 0.48 | 29.81 ± 0.34 | 31.49 ± 1.02 | 30.39 ± 0.27 | 20.70 ± 1.51 | 30.77 ± 0.55 |
| CIFAR100 with **Blend** Attack, **Poison** Accuracy | 0.15 | 46.72 ± 0.23 | 46.56 ± 0.26 | 46.59 ± 0.43 | 46.83 ± 1.00 | 42.15 ± 0.68 | 46.80 ± 0.70 |
| | 0.25 | 41.64 ± 1.54 | 40.60 ± 0.98 | 43.43 ± 0.59 | 40.02 ± 1.44 | 34.10 ± 1.60 | 39.30 ± 3.16 |
| | 0.35 | 31.18 ± 2.83 | 30.91 ± 2.02 | 35.84 ± 1.71 | 28.86 ± 2.82 | 25.36 ± 2.65 | 28.94 ± 3.49 |
| | 0.45 | 22.98 ± 1.18 | 23.37 ± 0.92 | 24.60 ± 2.19 | 22.16 ± 3.73 | 19.57 ± 1.64 | 24.17 ± 2.70 |
| CIFAR100 with **Blend** Attack, **Clean** Accuracy | 0.15 | 48.05 ± 0.33 | 47.83 ± 0.29 | 47.24 ± 0.65 | 48.43 ± 0.44 | 42.94 ± 0.55 | 48.37 ± 0.68 |
| | 0.25 | 44.85 ± 0.52 | 44.59 ± 0.31 | 44.17 ± 0.36 | 45.19 ± 0.26 | 36.18 ± 1.20 | 45.06 ± 0.18 |
| | 0.35 | 40.80 ± 0.56 | 40.08 ± 0.41 | 38.23 ± 0.80 | 41.84 ± 0.51 | 30.23 ± 1.25 | 41.18 ± 0.69 |
| | 0.45 | 35.32 ± 1.77 | 34.13 ± 1.13 | 27.33 ± 1.37 | 39.06 ± 0.45 | 24.22 ± 1.66 | 38.62 ± 1.30 |

Table 5: Ablation study on CIFAR100. $\epsilon$ is the corruption rate.

backdoor data poisoning attack, adding the adversarial training on existing algorithm could largely improve the robustness against backdoor attack.Both theoretical and empirical analysis show the effectiveness of our proposed meta algorithm.

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

# 7 APPENDIX

In this section, we provided proof of theorem and more discussion.

## 7.1 PROOF OF EQUATION 3

$$\frac{1}{n} \sum_{\mathbf{x} \in \mathbf{X}, \mathbf{y} \in \mathbf{Y}} [\mathcal{L}(f(\mathbf{x} + \mathbf{c}), \mathbf{y})] \leq \frac{1}{n} \sum_{\mathbf{x} \in \mathbf{X}^\epsilon, \mathbf{y} \in \mathbf{Y}} \phi_w(\mathbf{x}_i + c, \mathbf{y}) + \epsilon \tau L$$

let $\mathcal{G}$ denote the initially clean sample set (i.e. $(\mathbf{X}, \mathbf{Y})$), and $\mathcal{B}$ the corrupted sample set (*i.e.* the training set corrupted with a trigger whereas the labels are untouched). Let $\mathcal{R}$ denote the clean sample set which is replaced by the adversary (i.e. $\mathcal{R}$ is the subset of $\mathcal{G}$, and is replaced by $\mathcal{B}$, *i.e.* $\mathcal{G}' = \mathcal{G} \setminus \mathcal{R} \cup \mathcal{B} = (\mathbf{X}^\epsilon, \mathbf{Y})$), and let $\phi_w$ denote the function $\mathcal{L} \circ f$.

One can decompose the inner part of our mini-max objective in Equation 2 as the following,

$$\frac{1}{n} \sum_{\mathbf{x} \in \mathbf{X}, \mathbf{y} \in \mathbf{Y}} [\mathcal{L}(f(\mathbf{x} + \mathbf{c}), \mathbf{y})] = \frac{1}{n} \sum_{i \in \mathcal{G}' \setminus \mathcal{B}} \phi_w(\mathbf{x}_i + c, \mathbf{y}) + \frac{1}{n} \sum_{i \in \mathcal{R}} \phi_w(\mathbf{x}_i + c, \mathbf{y})$$

$$= \frac{1}{n} \sum_{i \in \mathcal{G}' \setminus \mathcal{B}} \phi_w(\mathbf{x}_i + c, \mathbf{y}) + \frac{1}{n} \sum_{i \in \mathcal{R}} \phi_w(\mathbf{x}_i + c, \mathbf{y}) + \frac{1}{n} \sum_{i \in \mathcal{B}} \phi_w(\mathbf{x}_i + \mathbf{t} + c, \mathbf{y}) - \frac{1}{n} \sum_{i \in \mathcal{B}} \phi_w(\mathbf{x}_i + \mathbf{t} + c, \mathbf{y})$$

$$= \frac{1}{n} \sum_{\mathbf{x} \in \mathbf{X}^\epsilon, \mathbf{y} \in \mathbf{Y}} \phi_w(\mathbf{x}_i + c, \mathbf{y}) + \left( \frac{1}{n} \sum_{i \in \mathcal{R}} \phi_w(\mathbf{x}_i + c, \mathbf{y}) - \frac{1}{n} \sum_{i \in \mathcal{B}} \phi_w(\mathbf{x}_i + c + \mathbf{t}, \mathbf{y}) \right)$$

$$\leq \frac{1}{n} \sum_{\mathbf{x} \in \mathbf{X}^\epsilon, \mathbf{y} \in \mathbf{Y}} \phi_w(\mathbf{x}_i + c, \mathbf{y}) + \left| \left( \frac{1}{n} \sum_{i \in \mathcal{R}} \phi_w(\mathbf{x}_i + c, \mathbf{y}) - \frac{1}{n} \sum_{i \in \mathcal{B}} \phi_w(\mathbf{x}_i + c + \mathbf{t}, \mathbf{y}) \right) \right|$$

$$\leq \frac{1}{n} \sum_{\mathbf{x} \in \mathbf{X}^\epsilon, \mathbf{y} \in \mathbf{Y}} \phi_w(\mathbf{x}_i + c, \mathbf{y}) + \epsilon L \|t\| \leq \frac{1}{n} \sum_{\mathbf{x} \in \mathbf{X}^\epsilon, \mathbf{y} \in \mathbf{Y}} \phi_w(\mathbf{x}_i + c, \mathbf{y}) + \epsilon \tau L,$$

## 7.2 PROOF OF THEOREM 1

**Theorem.** *Let* $\tilde{\mathcal{R}}_c^{emp}, \mathcal{R}_c^{emp}, \mathcal{R}_t, \epsilon, \tau$, *is defined as above. Assume the prior distribution of the network parameter* $\mathbf{w}$ *is* $\mathcal{N}(0, \sigma)$, *and the posterior distribution of parameter is* $\mathcal{N}(\mathbf{w}, \sigma)$ *is the posterior parameter distribution, where* $\mathbf{w}$ *is learned according to training data. Let* $k$ *to be the number of parameters,* $n$ *to be the sample size, assume the objective function* $\phi_\mathbf{w} = \mathcal{L} \circ f$ *is* $L_\phi$-*lipschitz smooth, then, with probability at least 1-$\delta$, we have:*

$$\mathcal{R}_t \leq \mathcal{R}_c^{emp} + L_\phi(2\tau + \epsilon\tau) + \sqrt{\frac{\frac{1}{4}k \log\left(1 + \frac{\|\boldsymbol{w}\|_2^2}{k\sigma^2}\right) + \frac{1}{4} + \log\frac{n}{\delta} + 2\log(6n + 3k)}{n - 1}}$$

Proof: we first decompose the gap as following

$$\mathcal{R}_t - \mathcal{R}_c^{emp} = (\mathcal{R}_t - \mathcal{R}_t^{emp}) + (\mathcal{R}_t^{emp} - \mathcal{R}_c^{emp}) \leq |(\mathcal{R}_t - \mathcal{R}_t^{emp})| + |(\mathcal{R}_t^{emp} - \mathcal{R}_c^{emp})|$$

We bound the second part first.

$$\mathcal{R}_t^{emp} - \mathcal{R}_c^{emp} \le \|\mathcal{R}_t^{emp} - \mathcal{R}_c^{emp}\|$$

$$= \frac{1}{n}\| \sum_{\mathbf{x}\in\mathbf{X}_r,\mathbf{y}\in\mathbf{Y}_r} [\phi(\mathbf{x}+\mathbf{t},\mathbf{y}) - \phi(\mathbf{x}+\mathbf{c},\mathbf{y})] + \left[ \sum_{\mathbf{x}\in\mathbf{X}_o,\mathbf{y}\in\mathbf{Y}_o} \phi(\mathbf{x}+\mathbf{t},\mathbf{y}) - \sum_{\mathbf{x}\in\mathbf{X}_b,\mathbf{y}\in\mathbf{Y}_o} \phi(\mathbf{x}+\mathbf{c},\mathbf{y}) \right] \|$$

$$\le \frac{1}{n}\| \sum_{\mathbf{x}\in\mathbf{X}_r,\mathbf{y}\in\mathbf{Y}_r} [\phi(\mathbf{x}+\mathbf{t},\mathbf{y}) - \phi(\mathbf{x}+\mathbf{c},\mathbf{y})] \| + \frac{1}{n}\| \sum_{\mathbf{x}\in\mathbf{X}_o,\mathbf{y}\in\mathbf{Y}_o} \phi(\mathbf{x}+\mathbf{t},\mathbf{y}) - \sum_{\mathbf{x}\in\mathbf{X}_b,\mathbf{y}\in\mathbf{Y}_o} \phi(\mathbf{x}+\mathbf{c},\mathbf{y})\|$$

$$\le (1-\epsilon)L_\phi\|\mathbf{t}-\mathbf{c}\| + \epsilon L_\phi\|\mathbf{t}-\mathbf{c}\| + L_\phi \max_{\mathbf{x}_o,\mathbf{x}_b} \|\mathbf{x}_o - \mathbf{x}_b\|$$

$$\le (1-\epsilon)L_\phi\|\mathbf{t}-\mathbf{c}\| + \epsilon L_\phi\|\mathbf{t}-\mathbf{c}\| + \epsilon L_\phi\|\mathbf{t}\|$$

$$= L_\phi\|\mathbf{t}-\mathbf{c}\| + \epsilon L_\phi\|\mathbf{t}\|$$

$$\le L_\phi 2\tau + \epsilon L_\phi\|t\|$$

$$\le L_\phi(2\tau + \epsilon\tau)$$

Now, we bound the second term. Note the second term is a typical gap term between empirical loss and generalization loss, and there are many approaches to bound this term like VC dimension. Since we aimed to focus the deep neural network, we follow the PAC-Bayes framework McAllester (1999) to analyze the generalization bound. Specifically, we use results from Foret et al. (2020), which gives $\sqrt{\dfrac{\frac{1}{4}k\log\left(1+\frac{\|\boldsymbol{w}\|_2^2}{k\sigma^2}\right)+\frac{1}{4}+\log\frac{n}{\delta}+2\log(6n+3k)}{n-1}}$ under the assumption of gaussian prior and posterior. The proof for this can be found in the appendix of Foret et al. (2020) (i.e. equation 13 on the paper).

### 7.3 DISCUSSION ABOUT LIPSCHITZ REGULARIZATION AND ADVERSARIAL TRAINING

As we could see from the above theorem that a small lipschitz constant could bring robustness against backdoor attack. In this section, we provided the reason why we claim adversarial training helps lipschitz regularization. The definition of lipschitz function is $\|f(\mathbf{x}) - f(\mathbf{y})\| \le L\|\mathbf{x} - \mathbf{y}\|, \forall \mathbf{x}, \mathbf{y}$. Since the lipschitz constant showes in the upper bound of the error, we would like to get the minimum lipschitz constant to tighten the bound. Follow (Terjék, 2019), the minimum lipschitz constant can be expressed as:

$$\|f\|_L = \sup_{x,y\in X; x\neq y} \frac{d_Y(f(x),f(y))}{d_X(x,y)}$$

rewrite $y$ as $x + c$, we would get

$$\|f\|_L = \sup_{x,x+r\in X; 0<d_X(x,x+c)} \frac{d_Y(f(x),f(x+r))}{d_X(x,x+r)}$$

Minimize above objective respect to function $f$ reduces to the adversarial learning.

$$\inf_f \|f\|_L = \inf_f \sup_{x,x+c\in X; 0<d_X(x,x+c)} \frac{d_Y(f(x),f(x+c))}{d_X(x,x+c)}$$

If we treat the denominator as a constant, then this is exactly the same as our minimax objective. More details can be found in (Terjék, 2019).

### 7.4 SUPPLEMENTARY EXPERIMENT RESULTS

We provided the experiment hyperparameters, and supplementary results for the experiment.

#### 7.4.1 EXPERIMENT HYPERPARAMETERS

We list the details of experiment in this section. All the methods use resnet-32 as the backbone network. AdamW is used as the opimitzer for all methods. The perturbation limit $\tau$ is set to be 0.05 for all methods requiring $\tau$. All methods are repeated for three different random seeds to calculate the standard deviation.

The trigger for badnet attack and blending attack can be found in figure 2.

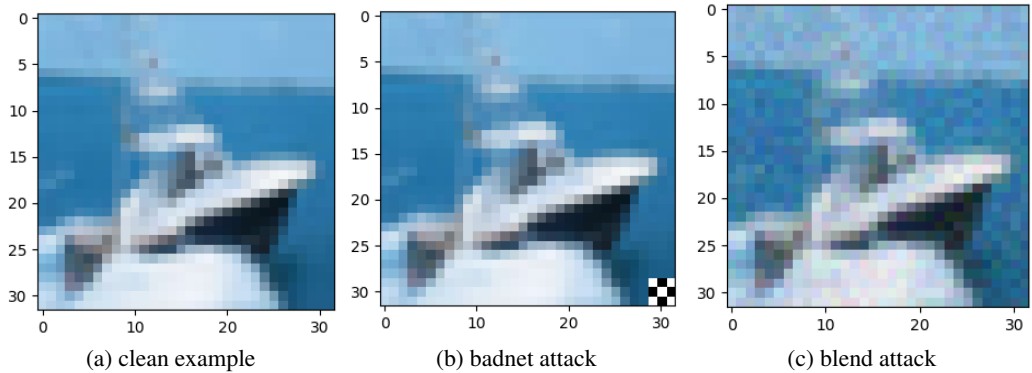

| (a) clean example | (b) badnet attack | (c) blend attack |

Figure 2: Example of clean and various poisoned samples

### 7.4.2 EXPERIMENT ON MNIST

We found interesting results on MNIST. In MNIST, we found adversarial training itself sometimes gives robustness to the backdoor attack. We hypothesis that this is because the MNIST is potentially a easier task than CIFAR. In here, we show the performance of adversarial training and PRL-AT on MNIST. The results can be found at table 6.

| Dataset | $\epsilon$ | AT | BootStrap | Bootstrap-AT | PRL | PRL-AT | SPL | SPL-AT |
|---|---|---|---|---|---|---|---|---|
| | | | | | **Backdoor Attack Defense Accuracy.** | | | |
| MNIST with **Patch** Attack, | 0.15 | $0.30 \pm 0.07$ | $0.04 \pm 0.01$ | $3.17 \pm 3.23$ | $97.96 \pm 0.21$ | $98.44 \pm 0.05$ | $59.24 \pm 33.30$ | $85.61 \pm 12.56$ |
| **Poison** Accuracy | 0.25 | $0.26 \pm 0.17$ | $0.04 \pm 0.02$ | $0.17 \pm 0.10$ | $89.91 \pm 7.53$ | $97.04 \pm 1.07$ | $25.25 \pm 1.38$ | $30.70 \pm 6.62$ |
| | 0.35 | $0.10 \pm 0.02$ | $0.08 \pm 0.06$ | $0.14 \pm 0.01$ | $77.91 \pm 10.41$ | $97.71 \pm 0.18$ | $13.54 \pm 0.76$ | $26.03 \pm 5.27$ |
| | 0.45 | $0.11 \pm 0.01$ | $0.04 \pm 0.02$ | $0.42 \pm 0.34$ | $43.42 \pm 11.66$ | $76.42 \pm 8.65$ | $12.85 \pm 1.90$ | $10.97 \pm 2.16$ |
| MNIST with **Patch** Attack, | 0.15 | $98.17 \pm 0.69$ | $99.49 \pm 0.05$ | $95.48 \pm 1.56$ | $98.08 \pm 0.26$ | $98.44 \pm 0.05$ | $93.23 \pm 4.72$ | $97.74 \pm 0.37$ |
| **Clean** Accuracy | 0.25 | $98.59 \pm 0.22$ | $99.48 \pm 0.07$ | $98.83 \pm 0.19$ | $97.46 \pm 0.07$ | $97.11 \pm 1.06$ | $86.98 \pm 0.72$ | $85.89 \pm 1.76$ |
| | 0.35 | $94.48 \pm 4.87$ | $99.48 \pm 0.04$ | $98.45 \pm 0.32$ | $97.40 \pm 0.46$ | $97.86 \pm 0.11$ | $73.09 \pm 4.34$ | $77.49 \pm 0.96$ |
| | 0.45 | $98.27 \pm 0.43$ | $99.42 \pm 0.02$ | $96.44 \pm 1.36$ | $75.69 \pm 0.99$ | $92.32 \pm 4.25$ | $60.58 \pm 1.90$ | $57.20 \pm 0.37$ |
| MNIST with **Blend** Attack, | 0.15 | $63.42 \pm 35.24$ | $0.04 \pm 0.01$ | $96.66 \pm 2.58$ | $96.81 \pm 1.30$ | $96.74 \pm 1.03$ | $97.43 \pm 0.13$ | $96.16 \pm 0.19$ |
| **Poison** Accuracy | 0.25 | $70.43 \pm 28.61$ | $0.04 \pm 0.01$ | $97.83 \pm 0.91$ | $77.68 \pm 20.34$ | $97.20 \pm 0.66$ | $6.43 \pm 1.27$ | $83.86 \pm 2.74$ |
| | 0.35 | $58.32 \pm 40.59$ | $0.05 \pm 0.03$ | $97.94 \pm 0.57$ | $78.79 \pm 17.74$ | $97.59 \pm 0.12$ | $11.05 \pm 2.70$ | $69.69 \pm 6.59$ |
| | 0.45 | $97.66 \pm 1.04$ | $0.03 \pm 0.03$ | $98.16 \pm 0.58$ | $27.18 \pm 19.53$ | $95.17 \pm 1.83$ | $4.49 \pm 1.08$ | $64.78 \pm 3.14$ |
| MNIST with **Blend** Attack, | 0.15 | $64.78 \pm 33.81$ | $99.44 \pm 0.02$ | $98.29 \pm 0.81$ | $97.93 \pm 0.25$ | $96.18 \pm 1.44$ | $97.30 \pm 0.22$ | $95.93 \pm 0.20$ |
| **Clean** Accuracy | 0.25 | $74.00 \pm 25.25$ | $99.46 \pm 0.05$ | $97.44 \pm 0.99$ | $97.30 \pm 0.63$ | $97.10 \pm 0.74$ | $77.75 \pm 0.74$ | $83.34 \pm 2.81$ |
| | 0.35 | $58.62 \pm 40.18$ | $99.44 \pm 0.02$ | $97.43 \pm 0.94$ | $96.25 \pm 1.84$ | $97.39 \pm 0.19$ | $72.41 \pm 3.80$ | $67.92 \pm 8.38$ |
| | 0.45 | $96.78 \pm 1.42$ | $99.42 \pm 0.06$ | $97.89 \pm 0.61$ | $76.47 \pm 8.66$ | $95.07 \pm 1.59$ | $63.63 \pm 4.47$ | $63.82 \pm 4.12$ |

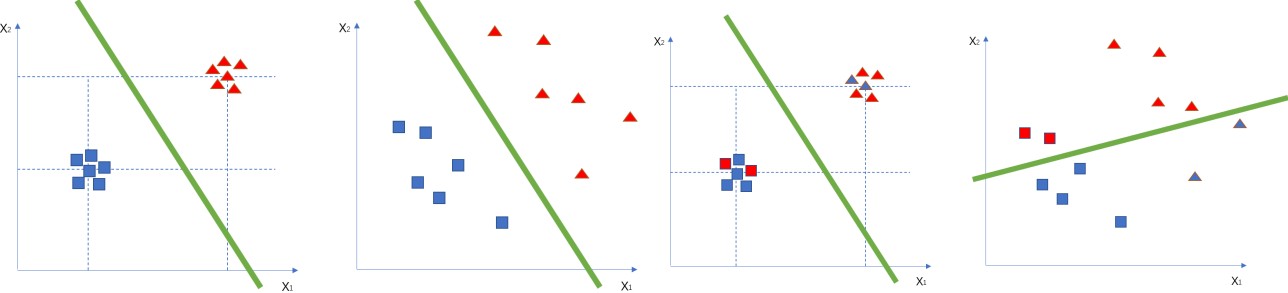

(a) clean data for binary feature value and continuous feature value (b) label flipping attack on both binary feature value and continuous feature value

Figure 3: Example of label flipping attack on both binary feature value and continuous feature value

As we can see for the MNIST, especially for the blend attack, the poison accuracy for adversarial training does show good performance with a large standard deviation. This is because that some random seeds works while some random seed failed. We hypothesis that this is because MNIST

| clean/poison accuracy | 0.15 | 0.25 | 0.35 | 0.45 |
|---|---|---|---|---|
| patch (PRL-AT) | 80.25/**80.15** | 78.22/**78.14** | 75.10/**75.04** | 58.90/**58.90** |
| patch(SpectralSign) | 80.32/35.90 | **80.40**/29.02 | 72.01/51.59 | 24.01/24.10 |
| patch(Fine-Pruning) | **80.34**/56.67 | 79.50/60.85 | **79.10**/56.84 | **78.73**/44.21 |
| blend (PRL-AT) | 67.97/68.48 | 71.28/71.87 | 74.12/**74.32** | 61.78/**54.19** |
| blend (SpectralSign) | **83.60**/70.74 | **81.23**/75.40 | 76.63/66.87 | 62.53/41.32 |
| blend(Fine-Pruning) | 79.53/34.38 | 79.32/13.94 | **78.28**/23.71 | **76.70**/16.36 |

Table 7: Comparison of averaged performance across three random seed with other baselines on CIFAR10. The numbers are clean accuracy/poison accuracy. Note fine-pruning used 5% clean data

dataset has almost binary feature value. When adding a small gaussian noise on feature x, the label flipping attack cannot change the decision boundary much. That is why the noisy label algorithm seems is not as important as the noisy label algorithm in CIFAR dataset. We plot a two dimensional toy example in figure 3 to illustrate label flipping attack on continuous features and binary features. As we can see in the figure that for the binary value feature, the label flipping attack is not easy to change the decision boundary too much while it can easily change the decision boundary in the continuous feature value scenario. However, this is a very rough conjecture for the reason why in MNIST, adversarial training sometimes works. We leave the investigation of this phenomenon in the future work.

## 7.5 COMPARISON AGAINST OTHER BACKDOOR DEFENSE ALGORITHM

In this section, we show further results against other backdoor defense algorithms. There are many existing backdoor defense algorithms. However, we found that a large fraction of those algorithms are either designed for single target attack (Liu et al., 2018) or requires clean data (Liu et al., 2018; Wang et al., 2019; Li et al., 2021). To investigate how our framework performs compared with the previous study, We compare against the following two baselines in a similar setting:

- spectral signature (Tran et al., 2018): filtering data by examining the score of projecting to singular vector.
- fine-pruning (Liu et al., 2018): prune the model by deleting non-activated neurons. Note this method uses 5% clean training data.

The results are in the table 7. As we can see, PRL-AT has higher poisoned accuracy against Spectral Signature and Fine-Pruning on most settings while the clean accuracy is still high, which indicates the effectiveness of our algorithm. With high corruption ratio, we find that the robustness of spectral signature and fine-pruning significantly decreases while PRL-AT still gives reasonable poison accuracy. Our theorem provides a principled way to design new defense algorithms by leveraging more knowledge from noisy label attacks. In this comparison, we only use PRL as the noisy label algorithm, and potentially, by using a more powerful noisy label algorithm (i.e. combination of multiple methods), it is possible to get higher poisoning accuracy.

## 7.6 SENSITIVITY ANALYSIS OF THE $\epsilon$

One interesting question to ask is how our algorithm performs without knowing the corruption ratio. In this section, we provided the worst-case result, in which we fix $\epsilon = 0.5$ to perform the noisy label algorithm. When the ground truth corruption ratio is higher than a half, it is impossible to learn any meaningful classier. We test the PRL-AT in the CIFAR10 on both badnet and blending attacks. The results are in table 8. As we can see, our algorithm provides robustness even use highly-overestimated estimated $\epsilon$ compared with the standard training results in table 2, which suggests our algorithm is not sensitive to the hyperparameter $\epsilon$.

## 7.7 DISCUSSION ABOUT NOISY LABEL ALGORITHM

One key question to ask for our framework is how to choose the noisy label algorithm. In terms of practice, we found PRL gives consistent robustness against both badnet and blending attacks on

| corruption ratio | PRL-AT (patch) | PRL-AT (blend) |
|---|---|---|
| 0.15 | 68.14/68.00 | 67.97/68.48 |
| 0.25 | 71.78/71.74 | 71.28/71.87 |
| 0.35 | 74.26/74.15 | 74.17/74.32 |
| 0.45 | 69.91/27.02 | 64.78/54.19 |

Table 8: PRL-AT top 1 averaged accuracy across three random seeds. The first number is the clean accuracy while the second number is the poisoned accuracy. The $\epsilon$ is fixed to be 0.5.

different settings. This might be because that PRL is designed for agnostic corrupted supervision, which is suitable for a variety of types of label noise attacks.

From a theoretical perspective, analyzing how different noisy label algorithm can minimize the first term of RHS in equation 6 depends on which noisy label algorithm to use. Here we give a rough analyze of PRL. PRL guarantees converging to the $\epsilon$-approximated stationary point, where $\epsilon$ is the corrupted ratio. Formally, we have the following corollary:

**Corollary 1** (Convergence of PRL to clean objective (Liu et al., 2021)). *Assuming the maximum clean gradient before loss layer has bounded operator norm:$\|W\|_{op} \leq C$, applying PRL to any $\epsilon$-fraction supervision corrupted data, yields $\min_{t \in [T]} \mathbb{E}\left(\|\nabla\phi(\mathbf{w}_t)\|\right) = \mathcal{O}(\epsilon\sqrt{q})$ for large enough $T$, where $q$ is the dimension of the supervision.*

More details can be found on the in Liu et al. (2021). According to above corollary, let $\mathbf{w}_{PRL}$ is the solution get by PRL algorithm, we can have $\|\nabla_{\mathbf{w}_{PRL}}\mathcal{R}_c^{emp}\| = \mathcal{O}(\epsilon)$ (i.e. assume $q$ is small). With Polyak-Lojasiewicz (PL) condition with some constant $\mu$ such that $\frac{1}{2}\|\nabla f(x)\| \geq \mu(f(x) - f^*)$ holds, we have $\mu(\mathcal{R}_c^{emp} - \mathcal{R}_c^{emp*}) \leq \frac{1}{2}\|\nabla_{\mathbf{w}_{PRL}}\mathcal{R}_c^{emp}\| = \mathcal{O}(\epsilon)$. For a highly-overparameterized deep neural network, the global optima $\mathcal{R}_c^{emp*}$ is usually 0. Thus, we can conclude that with PL condition, using PRL as the noisy label algorithm in our framework can guarantee $\mathcal{R}_c^{emp}$ can be minimized to the order $\frac{1}{\mu}\mathcal{O}(\epsilon)$. This yields the following proposition:

**Proposition 1.** *Let $\mathcal{R}_t, \epsilon, \tau$, is defined as above. Assume the prior distribution of the network parameter $\mathbf{w}$ is $\mathcal{N}(0, \sigma)$, and the posterior distribution of parameter is $\mathcal{N}(\mathbf{w}, \sigma)$ is the posterior parameter distribution, where $\mathbf{w}$ is learned according to training data. Let $k$ to be the number of parameters, $n$ to be the sample size, assume the objective function $\phi_{\mathbf{w}} = \mathcal{L} \circ f$ is $L_\phi$-lipschitz smooth and satisfying the PL condition, which is $\frac{1}{2}\|\nabla\phi_{\mathbf{w}}\| \geq \mu(\phi_{\mathbf{w}} - \phi_{\mathbf{w}^*})$. then, with the assumption of bounded operator norm of gradient before loss layer, we have with probability at least 1-$\delta$, by applying PRL-AT, we have:*

$$\mathcal{R}_t \leq \frac{1}{\mu}\mathcal{O}(\epsilon) + L_\phi(2\tau + \epsilon\tau) + \sqrt{\frac{\frac{1}{4}k\log\left(1 + \frac{\|\boldsymbol{w}\|_2^2}{k\sigma^2}\right) + \frac{1}{4} + \log\frac{n}{\delta} + 2\log(6n + 3k)}{n - 1}}$$

In general, considering $\phi$ is a deep neural network, we believe the first term is difficult to analyze without further assumption (i.e. PL condition). Nevertheless, empirical study shows that many noisy label algorithms can effectively minimize the first term loss, which motivates our method to treat those algorithms as a black-box algorithm.

