# OpenReview forum: "Defending Backdoor Data Poisoning Attacks by Using Noisy Label Defense Algorithm"
_ICLR.cc/2022/Conference — ICLR 2022 Submitted_

### Official Review · Reviewer_MBRR · 2021-11-02

**Correctness:** 3
**Technical Novelty And Significance:** 3
**Empirical Novelty And Significance:** 3
**Recommendation:** 6
**Confidence:** 4

**Main Review:**

Pros:
By adding noisy label to the training set and performing adversarial training at the same time, the noisy-label defense algorithm can be directly utilized to defense backdoor attacks. The method proposed is simple yet effective. The proof procedure is sufficient and details are provided.

Cons:
1. In Table 2, it is unclear why Bootstrap-AT not perform well with respect to the improvement on the model robustness on CIFAR10 with Patch Attack. The authors did not provide any discussion.

2. The computational cost of the method would be high since adding noisy label together with adversarial training takes much longer time. I suggest to provide a comparison and discussion on the times with other methods against backdoor attacks.

3. The organization of this paper should be improved. The analysis on experiments is short and incomplete.

4. There is a typo in your ‘Proof of Equation 3’. The range of the ∑ on the right-hand side of the first equation in the derivation should be i∈G\R.


**Summary Of The Paper:**

This paper aims to apply noisy-label defense algorithms to defend general backdoor attacks. The authors propose a meta-algorithm by adding noisy labels to the training set and optimizing outer minimization and inner maximization of the model by the noisy-label algorithm. Combined with adversarial training, the authors show their method could improve the robustness of models against backdoor attack.

**Summary Of The Review:**

The authors theoretically proved that noisy-label defense algorithms can be leveraged to against backdoor attacks, but the method seems not very innovative, and the computational cost would be high compared with other methods.

---

> ### Author Response · Authors · 2021-11-19
> **Response**
>
> We thank the reviewer for their valuable feedback and suggestions. Below are our point-to-point responses to your questions.
>
> Q1:
> In Table 2, it is unclear why Bootstrap-AT does not perform well with respect to the improvement on the model robustness on CIFAR10 with Patch Attack. The authors did not provide any discussion.
>
> We hypothesize that this is because the bootstrap algorithm is a more heuristic-based noisy label algorithm compared to SPL and PRL. In this paper, we treat noisy label algorithms as black-box algorithms, and analysis of how different noisy label algorithms can effectively minimize the first term in our theorem is left for future studies. This is saying that bootstrap-AT did not perform well with respect to improvement is because of the design of bootstrap instead of our framework. In general, we can see our framework did boost the robustness for all three noisy label algorithms in most settings.
>
> Q2:
> The computational cost of the method would be high since adding noisy label together with adversarial training takes much longer time. I suggest providing a comparison and discussion on the times with other methods against backdoor attacks
>
> Firstly, all three noisy label algorithms we choose are very efficient. For example, the PRL and SPL require sorting the loss/gradient norm on each backpropagation, which only requires $O(n \log n)$ extra complexity, where n is the mini-batch size.  For the bootstrap, it only requires adding a regularization to the loss, which does not require further computation.  For the adversarial training, we take one maximization step per iteration, thus for each iteration, our algorithm complexity is roughly twice compared against normal training, which is acceptable.
>
> Q3:
> The organization of this paper should be improved. The analysis of experiments is short and incomplete
> Thanks for the suggestion. We have added more discussion in the appendix.
>
> Q4:
> There is a typo in your ‘Proof of Equation 3’. The range of the ∑ on the right-hand side of the first equation in the derivation should be i∈G\R
> Thanks for pointing it out. We have fixed that typo in the proof.

---

### Official Review · Reviewer_62pH · 2021-11-03

**Correctness:** 2
**Technical Novelty And Significance:** 3
**Empirical Novelty And Significance:** 2
**Recommendation:** 5
**Confidence:** 3

**Main Review:**

### **Strengths**:

— The authors present a novel optimization based approach to defend against poisoning attacks. Theoretical justification is provided to motivate the loss and the effect they have on robustness.

— The proposed approach can be agnostic to the noisy label algorithm used during training, which makes it appealing to use with more advanced methods.

### **Concerns**:

— The experimental section is bit lacking. The proposed approach is not compared with previous defenses for noisy label attacks such as Spectral Signature defense.

— More advanced attacks where clean label is used  HTBA (arXiv:1910.00033 ) and Clean label backdoor attacks by Turner et al. have been presented which make the attack more effective. The authors do not consider such attacks in their settings and it would be interesting to see if the proposed work can be adapted for such scenarios.

— The organization of the work can be improved. The experiment section can include previous works for comparison and ablation studies along with hyperparameter related experiments can be considered in another section. This would make analyzing the experiments much easier.

— For the Blend attack, PRL based method performs poorly compared to SPL method, but PRL-AT outperforms SPL-AT. This is seen in multiple settings. The authors mention that filtering based algorithms are more easily boosted, but it is not intuitively clear why? A better noisy label algorithm should correspond to better performance irrespective of AT. Such a case makes it difficult to identify which part of the algorithm is responsible for the improvement.

— **Minor**: Some typos are present in the submitted version which can be corrected. Some of them include

"In the meantime, it is non-trival"- "non-trivial"

" we can approximate rvc as the following" - "we can approximate c as the following"

"resnet-32"  - "ResNet-32"

"since these two attacks does not require knowing any information about the model or training procedure" - "since these two attacks do not require any information about the model or training procedure".

**Summary Of The Paper:**

The proposed work presents a defense algorithm for backdoor poisoning attacks. The authors consider a noisy label setting where the attacker adds a trigger to certain samples of training data and changes their label. This mainly consists of two attacks, BadNets and Blending attack. The authors then come up with an objective to train the network in a noisy label setting such that networks become robust to the poisoning attacks.

**Summary Of The Review:**

The authors presenting an interesting approach towards defending against backdoor attacks involving noisy label algorithm. However, a more thorough comparison with previous defenses and other forms of attacks is required to understand the significance of the method. An optimization based approach towards defending against both clean and noisy label attacks would be interesting.

---

> ### Author Response · Authors · 2021-11-19
> **Response**
>
> We thank the reviewer for their valuable feedback and suggestions. Below are our point-to-point responses to your questions.
>
> Q1:
> Comparison with spectral signature
>
> Thanks for your suggestion. Please refer to the result from Table 7 in the appendix. We find our method shows comparable results against spectral signature when the corruption ratio is low and gives higher robustness when the corruption ratio is high.
>
> Q2:
> The organization of the work can be improved. The experiment section can include previous works for comparison and ablation studies along with hyperparameter-related experiments that can be considered in another section. This would make analyzing the experiments much easier.
>
> Thanks for your advice.  We added the sensitivity analysis in the appendix and will re-organize the paper as suggested.
>
> Q3:
> For the Blend attack, PRL based method performs poorly compared to the SPL method, but PRL-AT outperforms SPL-AT. This is seen in multiple settings. The authors mention that filtering-based algorithms are more easily boosted, but it is not intuitively clear why? A better noisy label algorithm should correspond to better performance irrespective of AT. Such a case makes it difficult to identify which part of the algorithm is responsible for the improvement
>
> A better noisy label algorithm should be able to achieve better performance against noisy label attacks. However, there is no guarantee that a better noisy label algorithm can perform better against backdoor attacks. That is why PRL performs worse than SPL against backdoor attacks. However, combined with adversarial training, we find PRL-AT does perform better than SPL-AT, which validates the results of our theorem.

---

> > ### Comment · Reviewer_62pH · 2021-11-30
> > **Thanks for the response**
> >
> > Thanks to the authors for providing additional experiments. It can be seen from Table 7 that PRL-AT outperforms spectral signatures on poison accuracy. However, it is still not clear if the proposed work can be adapted to more advanced clean label attacks. It is also not clear as to why AT helps more for certain noisy label algorithms (PRL) compared to others (SPL), even according to the theorem. For these reasons, I will keep my score unchanged.

---

### Official Review · Reviewer_tutw · 2021-11-05

**Correctness:** 3
**Technical Novelty And Significance:** 3
**Empirical Novelty And Significance:** 2
**Recommendation:** 3
**Confidence:** 4

**Main Review:**

The idea of using techniques for dealing with noisy labels for defending against backdoor data poisoning attacks looks interesting and promising. The motivation and the derivation of the algorithm seem reasonable, although the proposed algorithm has some parameters whose selection can be critical for the performance on the clean dataset and the backdoors.

I appreciate that the authors strived to provide a theoretical justification (Section 4.2) of the proposed algorithm. Thus, the authors propose a theorem that gives an upper bound for the risk given the empirical risk on the poisoned dataset with the clean labels (R_c^emp). The authors argue that the first term in the upper bound can be minimized by using a noisy label algorithm, but it is not clear what is the gap between the risk on the poisoned dataset with the flipped labels when using the noisy label algorithms. Can the authors provide some theoretical guarantee here? Perhaps the authors could shed a bit of light in this sense during the rebuttal.

My main concerns about the paper are on the experimental evaluation:
1) The results in Tables 1 and 2 shows that the proposed method has a significant negative impact on the performance of the algorithm on the clean dataset, which limits the applicability of the method.

2) The algorithms for mitigating the effect of noisy labels have parameters that assume the fraction of noisy labels that are present in the data. The influence of this parameter can be non-negligible and, a priori is not possible to foresee what is the fraction of poisoned training points for the backdoor attack. Similarly, the perturbation limit (\tau) used for the adversarial training in the proposed algorithm can also have an impact both on the performance on the clean dataset and on the reduction of the success of the backdoor attacks. For example, if the value of epsilon used for SPL or PRL is very large (e.g. 0.45) and there is no backdoor attack or the amount of poisons is very low (e.g. 1%) the performance on the clean dataset could be severely affected. Conversely, if epsilon is low and the fraction of poisons is high, the algorithm may not be able to defend against the backdoor attack. Thus, setting the value of the parameters of the algorithm is not trivial. In this sense, the authors should provide a more comprehensive analysis on the impact of the parameters on the accuracy both on the clean and the poison accuracy.

3) The experimental evaluation lacks comparison with other existing methods to defend against backdoors.

4) The scenarios with high levels of poisoning (e.g. 45%) are not really representative of a backdoor attack, where the objective of the attacker is to keep the performance of the resulting model when evaluated on the clean dataset. For large fractions of poisons, the performance on the clean dataset is severely affected as it can be observed in the tables with the results.

5) The experiments only consider CIFAR datasets. I think it would be convenient to include experiments in other benchmarks and, ideally, using other model architectures.


**Summary Of The Paper:**

The paper introduces an algorithm to defend against backdoor data poisoning attacks by leveraging noisy-label defense algorithms. Per se, noisy-label algorithms are not capable of defending against backdoors. However, the authors propose to use these techniques in combination with their proposed algorithm for adversarial training to reduce the effectiveness of backdoor data poisoning attacks.

**Summary Of The Review:**

The motivation for the proposed algorithm is interesting and promising. However, the empirical evaluation raises important concerns about the applicability and usefulness of the method to defend against backdoor poisoning attacks. On one side, the drop in performance on regular inputs when using the proposed technique is non-negligible and, on the other hand, the selection of the parameters of this algorithm can also have a significant impact both on the performance on the clean and the poisoned dataset (and it is not clear how to set this parameters properly). The experimental evaluation should be more comprehensive.

---

> ### Author Response · Authors · 2021-11-19
> **Response**
>
> We thank the reviewer for their valuable feedback and suggestions. Below are our point-to-point responses to your questions.
>
> Q1:
> theoretical guarantee of noisy label algorithm to minimize the first term in equation (6):
>
> Please refer to section 7.7 in the appendix to see the analysis of the noisy label algorithm. We provided a new upper bound specifically for PRL-AT in the updated version.
>
> Our proposed algorithmic framework treats the noisy label algorithm as a black-box algorithm, thus analyzing how noisy label algorithms minimize the first term is beyond the scope of the paper. In particular, the strategy for analyzing the first term depends on the choice of the noisy label algorithms. Considering the $\phi$ function is usually highly non-convex, it will be hard to guarantee that we could minimize the first term to achieve global minima if we do not add further assumptions to the function $\phi$. However, PRL has been proven to converge to the epsilon-approximated stationary point of the first term. If we add further assumptions (i.e. Polyak-Lojasiewicz condition), then we can see that the first term could be well bounded by using the PRL algorithm. However, in general, considering $\phi$ is a deep neural network in the paper, we believe the first term is difficult to analyze given the current assumptions. Nevertheless, empirical study shows that noisy label algorithms can effectively minimize the first term loss, which motivates our method to treat those algorithms as a black-box algorithm.
>
> Q2:
> the proposed method would degrade the performance on the clean dataset
>
> There are two reasons for this phenomenon. First, The tradeoff between normal generalization (clean accuracy) and worst-case generalization (poison accuracy) widely exists and how to identify the optimal tradeoff remains an open problem for deep neural networks.  To the best of our knowledge, no existing approaches achieve robustness without sacrificing clean accuracy. For example, in the Neural Attention Distillation (Li et. al. 2021), even using a small set of clean training data, the accuracy still drops compared to standard training. Thus, we are studying how to achieve robustness while keeping normal generalization performance as good as possible.  Secondly, in our paper, we use a large corruption rate, which decreases the useful sample size. This could be another reason why performance on clean datasets decreases.
>
> Q3: The effect of $\epsilon$
> Firstly, most of the previous literature assumes knowledge about $\epsilon$ (i.e. Spectral signature). In fact, almost all detection-based defense algorithms assume prior knowledge similar or equivalent to  $\epsilon$.
>
> To address your concern, we rerun the PRL-AT on CIFAR10 blend attack for 15%, 25%, 35%, and 45% corruption ratio and set the $\epsilon$ to be 0.5 for all the settings. When we assume no knowledge of the corruption ratio, this is the last thing we could do since if the corruption rate is bigger than 50%, it is impossible to learn meaningful classifiers. We attached the results in Table 8 of the appendix.
>
> We see that except when the corruption ratio is high on patch attack (i.e. 45% corruption ratio), even if we set the epsilon always to be 0.5, our algorithm still showed competitive performance. The relatively bad performance on the 15% corruption ratio is due to highly overestimated epsilon.
>
> Q4:lacks comparison against other baselines
>
> Please refer to our response to reviewer 1 (Cons 1).
>
> Q5: high level poisoning are not representative of a backdoor attack
>
> We agree that high levels of poisoning (i.e. 45%) are not representative of a backdoor attack, and we would like to emphasize that even under the extreme levels, the proposed framework still produces promising results. Specifically, our results show that the setting of a 35% corruption ratio still has good clean accuracy while most previous papers studied corruption ratios smaller than 10%. Also, our main goal of this paper is to connect label flipping attack and backdoor attack, and in label flipping attack, 45% corruption ratio is a common setting in studying noisy labels.

---

> > ### Comment · Reviewer_tutw · 2021-11-22
> > **Comments after rebuttal**
> >
> > Thank you very much for your responses and for the clarification about the theoretical aspects.
> >
> > I have read carefully your comments as well as the other reviewers' commments. I think that most of us agree on the value of the algorithm proposed and the theoretical contribution. However, the empirical evaluation needs improvement and, possibly, requires a different organization and a more comprehensive analysis. In this sense, the section with the experiments is not at the same level that the previous sections in the paper.
> >
> > Thus, I'm keeping my score, but I'd like to encourage the authors to review and improve the experiments and resubmit to another conference. I think that the paper has potential, but it still requires some more work.

---

### Official Review · Reviewer_2ape · 2021-11-07

**Correctness:** 4
**Technical Novelty And Significance:** 3
**Empirical Novelty And Significance:** 3
**Recommendation:** 5
**Confidence:** 4

**Main Review:**

Pros:
1. The paper is very well written and I really enjoyed reading it. Through a sequence of steps, the authors explained how they arrived at their final formulation and did a nice job in sketching the implication of their theoretical result.
2. The connection between the noisy-label attack and the backdoor attack has been well explained and later well-exploited to arrive at the formulation and also the AT-based minimax optimization.
3. For evaluation, the authors chose three noisy-label defenses (SPL, PRL, and Bootstrap) and demonstrated the efficacy of their meta-algorithms (SPL-AT, PRL-AT, Bootstrap-AT) in improving the defense accuracy on the CIFAR-10/CIFAR-100 datasets for two different attack settings.

Cons:
1. One of the main issues I have is with the evaluation. The authors only provide a comparison with AT, and the noisy-label defenses, however, they do not provide any evidence as to how their algorithm compares to other poisoning defenses. In the absence of that comparison, it is difficult to assess the benefits of the defense proposed in this paper.

2. Some insight about which noisy-label algorithm is better suited for the optimization would be beneficial. While the tables (Tables 2 and 3, and the ones with ablation results) do provide defense accuracies but it would be nice if the authors can provide any more insight if one defense is preferred over others. This also goes back to the assumption on the existence of at least one noisy-label defense that the authors mention in Pg. 4 (after Assumption 2). Can we use a mixture of defenses for the inner and outer optimizations?

3. The authors mention that their technique is more beneficial in the large-scale poisoning regime ($\epsilon=0.45$). In this regard, the numbers in Table 2 (for example) when $\epsilon=0.45$ are interesting. Often plain AT provides a strong baseline (better than some of the re-inforced variants). Also, there is no real consistency between which of the three re-inforced defenses would do better for attacks. This links back to my earlier point about providing more insights into the choice of the noisy-label defense.

Clarifications:
1. I could not understand the clean accuracy numbers in the evaluation. Maybe, I am missing something here, but I was expecting that it would correspond to the $\epsilon=0$ setting. Does the 'Standard' column reflect the accuracy over the $(1-\epsilon)$ fraction of non-poisoned data?

2. In Section 4.2, it would be great if the authors can clarify why $\tilde{\mathcal{R}}^{emp}_t$ is not compared with $\mathcal{R}_t$?

3. The approximation from (4) through the Taylor Series expansion on to (5), especially the use of FGSM for (5) can be explained a little better.

4. Typos:
a. "but also can has" -- pg 2 (last sentence)
b. In Sec 4, the dataset is denoted as $(X^{\epsilon}, Y^{\epsilon})$, however, in the previous section they have been defined as  $(X_{\epsilon}, Y_{\epsilon})$. -- pg 4
c. 'rvc' (maybe a latex equation label) is not defined (just above (5)) -- pg 5

**Summary Of The Paper:**

The authors propose a defense against backdoor data poisoning attacks by leveraging (reinforcing) existing defenses against noisy-label attacks.

**Summary Of The Review:**

Please see the detailed comments in the main review section.

---

> ### Author Response · Authors · 2021-11-19
> **Response**
>
> Thanks for the review and suggestions. We added a few sections in the appendix and they are highlighted in red. Below are our responses to your questions:
>
> Cons1:
> Lacks of comparisons to other poisoning defenses
>
> Answer:
> As suggested, We added results of two baselines in a similar setting: The first baseline is the spectral signature (Tran et. al., 2018), which uses robust mean estimation algorithms to remove the corrupted data. The second baseline is the fine-pruning method (Liu et. al., 2018), and we use 5% extra clean data for fine-pruning. We attached the result in Table 7 in the appendix. We found that our framework performs better than the two baselines, especially with high corruption ratios.
>
> We would like to point out that we do not claim that existing implementations of our framework outperform all other backdoor defense algorithms in all settings. Our main contribution is to present a novel connection between noisy label attacks and backdoor attacks, which provides a principled way to design new defense algorithms by leveraging more knowledge from noisy label attacks. The framework can take advantage of future advances of noisy label attacks and directly transform them for backdoor defense.
>
> We also noted that the proposed framework is designed for the challenging problem setting that no cleaning data is available. This, unfortunately, has ruled out many baselines approaches for backdoor defense since they require a small set of clean training data (for example, fine-tuning based method (Liu et. al., 2018), neural cleanses method (Wang et. al., 2019), Neural Attention Distillation (Li et. al. 2021)) while in our paper, we only have access to noisy data with epsilon-backdoored attacks without information about which data points are being attacked. As a result, it will not be a fair comparison. Also, methods in a similar spirit as neural cleanse are not designed for multi-target attacks.
>
> Cons2&Cons3:
> Lacks discussion about the noisy label algorithm
>
> Answer:
> We added a detailed discussion about the choice of noisy label algorithm to section 7.7 in the appendix.
> In short, we recommend choosing PRL as the noisy label algorithm, since PRL is designed for agnostic corrupted supervision and can handle a variety of types of label noise.
> Our theorem 1 indicates that a noisy-label algorithm of higher performance leads to better defense against backdoor attacks. In the submission, we see that for the filtering method, PRL-AT shows better robustness than SPL-AT for the backdoor attack in most settings. This suggests PRL is a better noisy label algorithm compared with SPL, which is also illustrated in the PRL paper (Liu et al. 2021).  As for Bootstrap, it is a consistency-based method and it is more heuristic-driven. The results for Bootstrap algorithms are somewhat mixed; sometimes they work (CIFAR100 patch poisoning) and sometimes do not (CIFAR10 patch poisoning).
>
> It is difficult to claim that a noisy label algorithm consistently outperforms other algorithms in all settings. However, our theoretical findings revealed the key idea that we can further optimize the noisy label algorithms to obtain improved robustness against backdoor attacks. One promising future work is to consider a mixture of defenses for the inner and outer optimizations in this proposed framework.
>
> Clarifications:
> 1. I could not understand the clean accuracy numbers in the evaluation. Maybe, I am missing something here, but I was expecting that it would correspond to $\epsilon=0$ the setting. Does the 'Standard' column reflect the accuracy over the $1-\epsilon$ fraction of non-poisoned data?
>
> Clean accuracy means we are still training on the $\epsilon$-backdoored data but testing on non-backdoored data (i.e. test data does not include any triggers). Standard training means we train the DNN in the normal way using the corrupted, $\epsilon$-backdoored data without adding any defense. As seen in the Standard column in our performance tables, without adding triggers, the model still performs well (i.e. high clean accuracy) while if the trigger is added, the model performs poorly (i.e. low poison accuracy)
>
> 2. In Section 4.2, it would be great if the authors can clarify why $\tilde{R_t^{emp}}$ is not compared with R_t?
> We could also use $\tilde{R_t^{emp}}$. To do so, we could just add $\tilde{R_t^{emp}}$ and minus $\tilde{R_t^{emp}}$ on the RHS, then applying the triangle inequality would yield similar results. In this case, the RHS will become the empirical loss over epsilon backdoored data plus the difference between $R_t^{emp}$ and $\tilde{R_t^{emp}}$, which should also be minimized by some robust noisy label algorithm (i.e. a robust noisy label algorithm should make this difference to be small). We use $R_t^{emp}$ since we think the current form is easier to understand. In its current form, the first term is directly a noisy label loss, which shows the necessity to use a noisy label algorithm.

---

### Author Response · Authors · 2021-11-30
**A summarized response to all reviewers and the AC panel**

Dear reviewers and AC panel,
We appreciate all reviewers for reviewing our paper and leaving valuable comments!
With all respect, we think some reviewers are biased due to some misunderstandings about the contribution of our paper.
We have tried our best to address his/her concerns but we got very limited comments from Reviewers until now (the last day of the discussion period).
Due to the situation, we would like to bring the major concerns and our summarized responses to the AC's attention.

Our main contribution is to present a novel connection between noisy label attacks and backdoor attacks, which provides a principled way to design new defense algorithms by leveraging more knowledge from noisy label attacks. The framework can take advantage of future advances of noisy label attacks and directly transform them for backdoor defense. Thus, we mainly focused on studying whether our framework can successfully transfer different kinds of noisy label algorithm to robust algorithms against backdoor attacks. All of the reviewers argue we lack empirical comparison with other backdoor defense algorithm baselines. We would like to clarify that firstly, the main contribution of the paper is a meta-algorithm, which is a framework instead of a specific algorithm, and our experiment is designed to validate the correctness of our meta-algorithm instead of the superior performance of one special case of our algorithm against other baselines. Our empirical study shows that our meta-algorithm does successfully defend against the backdoor attack by using noisy label algorithms.  Secondly, we evaluate our meta-algorithm in a very challenging setting compared to the existing algorithm. Most of the existing papers perform experiments with single target attack, low corruption ratio (i.e. less than 10%), a small set of clean data while our meta-algorithm uses none of the above assumptions.  At last, even though, we added new experiment results against other baselines as requested. The results show our algorithm gives comparable results against those baselines even under unfair settings. Unfortunately, no further feedback for this is given.


Also, most reviewers request us to provide further discussion about how to choose the noisy label algorithms. Though we added further discussion during rebuttal, giving both empirical and theoretical analysis, we get no further feedback for this.  A similar problem is the sensitivity analysis. We also added the sensitivity analysis as requested during rebuttal, none of the reviewers gives specific feedback for it.

---

### Decision · Program_Chairs · 2022-01-20

**Decision:**

Reject

**Comment:**

The paper proposes a new defense against backdoor attacks utilizing an improved version of defenses against noisy label attacks. The connection between these two problems is interesting and novel, which is acknowledge by all reviewers. The main drawback of the paper is, however, its experimental evaluation. The experiments are carried out only on one benchmark and the considered attack ratios are indicative for indiscriminative poisoning attacks rather than targeted backdoors. The AC addressed the summarized response and is not convinced that the reviews are biased. Despite the scarce response of the reviewers to the author rebuttal, the limitations of the experimental evaluation seem to persist in the revised version of the paper. While acknowledging the novelty and the overall good quality of the paper, the weakness of its experimental evaluation puts at in the position marginally below the acceptance threshold. The AC encourages the authors to revise the paper and improve on the pointed out weaknesses and is confident that this work will be well accepted by the scientific community.